# CTODS: Polynomial-Time Construction of Hypergraphs via Constrained Overlapping Densest Subgraphs for Enhanced Neural Network Performance

## Abstract

The fundamental challenge of constructing meaningful hyperedges from graph structures has limited the effectiveness of hypergraph neural networks in capturing complex relational patterns. We present CTODS (Constrained Top-$K$ Overlapping Densest Subgraphs), a theoretically grounded polynomial-time algorithm that transforms graphs into hypergraphs by systematically identifying overlapping dense subgraphs as hyperedges. The algorithm achieves computational efficiency with $O(K \cdot m \log n)$ time complexity and $O(n + m)$ space requirements while ensuring connectivity-enforced subgraph discovery through a principled distance function that controls overlap. Our adaptive parameter optimization framework enables robust performance across diverse network topologies, from citation networks to geometric structures. Extensive empirical validation across eight benchmark datasets demonstrates the method's superiority, achieving consistent improvements over existing approaches.

## 1 Introduction

In many real-world applications, Higher-Order Interactions (HOIs) are pervasive. HOIs reveal structural patterns unobserved in their pairwise counterparts and inform network dynamics (Kim et al., 2024; Jiang et al., 2019). Hypergraphs have recently attracted considerable attention across multiple domains, such as recommendation systems (Xia et al., 2022), social network analysis (Sun et al., 2023), cross-modal retrieval (Li et al., 2024b), and bioinformatics (Murgas et al., 2022). Formally speaking, hypergraphs express higher-order networks or networks of HOIs, where nodes and hyperedges, respectively, represent sets of entities and their HOIs.

In contrast to an edge connecting only two nodes in pairwise graphs, a hyperedge can connect an arbitrary number of nodes, offering hypergraphs advantages in their descriptive power. Graphs become hypergraphs when additional information in a graph groups entity nodes together into sets. The expressiveness and flexibility of hypergraphs have made them a powerful structure for modeling higher-order networks across various domains (Liao et al., 2021; Huang & Yang, 2021; Cai et al., 2022a). Recent advancements in hypergraph neural networks (HGNNs) (Bai et al., 2021; Feng et al., 2019; Li et al., 2025) have shown promise in modeling such interactions, yet defining meaningful hyperedges from raw graph structures remains a significant challenge. Standard hyperedge generation methods, such as k-nearest neighbors (Zhang et al., 2022) or k-hop neighborhoods (Li et al., 2024c), frequently fail to adequately capture truly cohesive higher-order relationships or become computationally inefficient and noisy, highlighting a crucial gap in current methodologies (Serviansky et al., 2020; Tang et al., 2023; Zhang et al., 2022).

To address these limitations, our work leverages a complementary structural approach focused on densest subgraphs which defines as cohesive, highly interconnected vertex subsets. This approach is particularly advantageous because both subgraphs and hyperedges are fundamentally vertex subsets (Li et al., 2024a), creating a natural structural alignment that facilitates meaningful hypergraph construction. Densest subgraphs naturally capture HOIs by identifying cohesive groups of vertices that are densely connected, closely mirroring real-world overlapping community patterns, which

are prevalent yet often inadequately modeled (Lanciano et al., 2024; Contisciani et al., 2022b; Lee et al., 2021a). Unlike classical approaches that primarily identify disjoint dense subgraphs, we propose a novel formulation known as Constrained Top-$k$ Overlapping Densest Subgraphs (CTODS), explicitly allowing controlled overlap among identified dense groups. This approach aligns with the practical observation that real-world vertices commonly belong to multiple communities, thus enabling more nuanced and realistic modeling of higher-order structures where each dense subgraph can directly serve as a hyperedge in the constructed hypergraph.

The main contributions of this paper are summarized as follows:

- Introduce CTODS, a polynomial-time algorithm that transforms graphs into hypergraphs by systematically identifying overlapping dense subgraphs as hyperedges, achieving $O(K \cdot m \log n)$ time complexity with strict connectivity and size constraints.

- Establish the theoretical foundation for using densest subgraphs as hyperedges, demonstrating that both are fundamentally vertex subsets, creating natural structural alignment that facilitates principled hypergraph construction while enabling controlled overlap through a distance-based diversity function.

- Conduct extensive experiments across eight diverse benchmark datasets, demonstrating consistent performance improvements of 1-3% over existing hypergraph neural networks and traditional graph neural networks, with particular success on citation networks, social networks, and geometric structures.

- Develop an adaptive parameter optimization approach that automatically adjusts the density-diversity trade-off parameter $\lambda$ based on graph structural properties, enabling robust performance across different network topologies without manual tuning.

## 2 RELATED WORK

Hypergraphs were introduced as an extension to traditional graph theory, allowing hyperedges to connect any number of vertices (Berge, 1973). They define hyperedges as a family of subsets of nodes, which lays the groundwork for our strategy to define and create hyperedges and, subsequently, hypergraphs.

Recent advancements in hypergraph neural networks (HGNNs) have demonstrated superior performance by extending the capabilities of graph neural networks to high-order relationships represented in terms of hypergraphs(Li et al., 2025; Feng et al., 2019; Li et al., 2024b). The authors of Bai et al. (2021) introduced hypergraph convolution and hypergraph attention mechanisms, which define a convolutional layer capable of handling tuple-wise relationships while ensuring flexible and discriminative learning. Similarly, the work of Feng et al. (2019) proposed the first hypergraph neural network (HGNN), which formalized a spectral-based approach to hypergraph convolution.

The work of Feng et al. (2019) introduces approaches like $Epair$ and $Ehop$ to transform graph structures into hypergraphs, by creating pairwise hyperedges or $k$-hop neighborhood groups, respectively. While $Epair$ effectively models low-order correlations, it lacks the ability to capture rich, multi-node relationships critical for HOIs modeling. On the other hand, $Ehop$ extends beyond pairwise relationships by expanding the neighborhood radius, although this often results in computational inefficiency, noisy correlations, and static hyperedge structures that are predefined and lack adaptability to task-specific data. Several methods have emerged to enhance graph-to-hypergraph construction. Chien et al. (2022) proposed the AllSet framework, allowing flexible multiset functions over hyperedges. While expressive, it assumes a fixed hypergraph structure, limiting adaptability. Cai et al. (2022b) addressed this limitation with Hypergraph Structure Learning (HSL), which dynamically updates hyperedges during training. HSL improves performance by tailoring hypergraph topology to the task but incurs extra computational cost and risks overfitting. Zhou et al. (2023) advanced this further with the Totally Dynamic Hypergraph Neural Network (TDHNN), which learns both the structure and number of hyperedges.

Related to the above, the problem of finding dense subgraphs and their variations has been extensively studied. If density is defined as the average degree of the subgraph, the problem becomes polynomial-time solvable (Niu et al., 2025). This was observed early on by Goldberg (1984), who designed an algorithm based on a transformation to the minimum-cut problem. For the same prob-

lem, (Asahiro et al., 2000) and Charikar (2000) provided a greedy linear-time 2-approximation algorithm, making the problem tractable for very large datasets. Previous algorithms, however, did not consider any constraint on the size of the densest subgraphs. All of the works discussed above focus on the problem of finding a single densest subgraph. Seeking to find the Top-$k$ densest subgraphs is a much less studied problem. The authors of Balalau et al. (2024) were the first to study the problem formally, though the problem formulation in this work differs significantly from ours, as they impose a hard constraint on the overlap when generating hyperedges, limiting the flexibility of their model in capturing overlapping communities.

The works of (Dondi et al., 2021; Galbrun et al., 2016; Zhou et al., 2024) densest subgraphs face scalability challenges, particularly for large-scale graphs and higher values of $k$. Moreover, their focus on graph-based structures does not extend to hypergraph-specific insights, where HOIs can be modeled through hyperedges. While other dense subgraph definitions exist, such as k-core, k-truss, densest subgraphs offer unique advantages for hypergraph construction (Khaouid et al., 2015; Wang & Cheng, 2012). K-core decomposition identifies nested structures but doesn't optimize for density, often resulting in subgraphs with varying quality (Chu et al., 2024; Zhang et al., 2025). K-truss focuses on triangle density but may miss broader structural patterns (Qin et al., 2025). In contrast, densest subgraphs directly optimize for the most cohesive vertex subsets, ensuring that each hyperedge represents a meaningful, well-connected group of nodes.

## 3 PRELIMINARIES

### 3.1 HYPERGRAPHS

Let $\mathcal{V}$ denote a finite set of elements, nodes, or objects, which we formally call "vertices". Let $\mathcal{E}$ be a family of subsets $e$ of $\mathcal{V}$ such that $\bigcup_{e \in \mathcal{E}} e = \mathcal{V}$. Then, we call $\mathcal{G}_h = (\mathcal{V}, \mathcal{E})$ a hypergraph with the vertex set $\mathcal{V}$ and the hyperedge set $\mathcal{E}$. A hyperedge containing just two vertices is a simple graph edge. A weighted hypergraph is one such graph that has a positive number $w(e)$ associated with each hyperedge $e$, showing the importance of the connections inside a hyperedge, called the weight of hyperedge $e$. Denote a weighted hypergraph as $\mathcal{G}_h = (\mathcal{V}, \mathcal{E}, \mathbf{W})$, where $\mathbf{W}$ is a diagonal matrix representing the weights of the hyperedges (Devine et al., 2006). Furthermore, we introduce a hypergraph with constraints on the hyperedge sizes as follows:

$$\mathcal{G}_h^{(\alpha, \beta)} = (\mathcal{V}, \mathcal{E}),$$

where each hyperedge $e \in \mathcal{E}$ satisfies the constraint $\alpha \leq |e| \leq \beta$, with $|e|$ denoting the number of vertices in hyperedge $e$. Later in this paper, we show why we need constraints on the hyperedge sizes, though for now, let us discuss the generic hypergraph.

A hypergraph $\mathcal{G}_h$ can be represented by a $|\mathcal{V}| \times |\mathcal{E}|$ matrix $\mathbf{H}$ with entries $h(v, e) = 1$ if $v \in e$ and 0 otherwise (Schölkopf et al., 2007). This is called the incidence matrix of $\mathcal{G}_h$. Then for a vertex $v \in \mathcal{V}$, its vertex degree is defined as

$$d(v) = \sum_{e \in \mathcal{E}} w(e) \mathbf{H}(v, e).$$

For a hyperedge $e \in \mathcal{E}$, its edge degree is defined as

$$d(e) = \sum_{v \in \mathcal{V}} \mathbf{H}(v, e).$$

$\mathbf{D}_v$ and $\mathbf{D}_e$ denote the diagonal matrices of vertex degrees and edge degrees, respectively. The initial feature set for each vertex is denoted as $X_0 = \{x_0^1, x_0^2, \ldots, x_0^N\}$ for $x_0^i \in \mathbb{R}^{C_0}$, where $C_0$ is the dimension of the feature.

The adjacency matrix $\mathbf{A}$ of hypergraph $\mathcal{G}_h$ is defined as:

$$\mathbf{A} = \mathbf{H}\mathbf{W}\mathbf{H}^T - \mathbf{D}_v,$$

where $\mathbf{H}^T$ is the transpose of $\mathbf{H}$.

After defining a hypergraph formally, we discuss efficient ways of creating hypergraphs that are suitable for HGNNs. The primary reason for choosing hypergraphs over graphs is HOIs, which

describe the underlying relationships. Considering the main purpose of hypergraphs, a *good* hyperedge should cover a subset of vertices that are related directly and indirectly to enhance the original structure of a graph and be able to encode the HOIs. If the hypergraph generation step is based on a graph structure, it can be seen as enriching the information of a graph that goes beyond the limitation of the graphs. The newly learned hypergraph structure serves as the input of HGNNs to learn better hyperedge and node representations.

# 4 METHODOLOGY

## 4.1 GRAPH DENSITY

Most recent works define the density of a graph $\mathcal{G} = (\mathcal{V}, \mathcal{E}')$ as follows:

$$\text{dens}(\mathcal{G}) = \frac{|\mathcal{E}'|}{|\mathcal{V}|}, \tag{1}$$

since it directly depends on the size of the vertex set ($|\mathcal{V}|$) and the edge set ($|\mathcal{E}'|$), both of which can be efficiently enumerated and computed in $O(|\mathcal{V}| + |\mathcal{E}'|)$ time. This makes the problem of finding the densest subgraph solvable in polynomial time (Gregory, 2007). Note that $\text{dens}(\mathcal{G})$ is half of the average degree of $\mathcal{G}$. If the graph $\mathcal{G} = (\mathcal{V}, \mathcal{E}')$ is known from the context and we are given a subset of vertices $\mathcal{L} \subseteq \mathcal{V}$, we define the density of $\mathcal{L}$ as follows:

$$\text{dens}(\mathcal{L}) = \text{dens}(\mathcal{G}(\mathcal{L})) \tag{2}$$

That is, the density of the induced subgraph $\mathcal{G}(\mathcal{L})$, where $\mathcal{G}(\mathcal{L})$ is the subgraph induced by the vertices in $\mathcal{L}$ and all edges in $\mathcal{E}'$ that connect vertices in $\mathcal{L}$. Additionally, we refer to the set $X \subseteq \mathcal{V}$ that maximizes $\text{dens}(X)$ as the densest subgraph of $\mathcal{G}$. Given a graph $\mathcal{G} = (\mathcal{V}, \mathcal{E}')$ and a collection of subsets of vertices $L = \{\mathcal{L}_1, \ldots, \mathcal{L}_k\}$, where $\mathcal{L}_i \subseteq \mathcal{V}$, We define the density of the collection $\mathcal{L}$ as the sum of the densities of the individual subsets, namely:

$$\text{dens}(\mathcal{L}) = \sum_{i=1}^{k} \text{dens}(\mathcal{L}_i). \tag{3}$$

Given a graph $\mathcal{G} = (\mathcal{V}, \mathcal{E}')$, and a subset $\mathcal{L} \subseteq \mathcal{V}$, we denote by $\mathcal{G}[\mathcal{L}]$ the subgraph of $\mathcal{G}$ induced by $L$; more formally $\mathcal{G}[\mathcal{L}] = (\mathcal{L}, \mathcal{E}'(\mathcal{L}))$, where $\mathcal{E}'(\mathcal{L})$ is defined as follows (Dondi et al., 2021) :

$$\mathcal{E}'(\mathcal{L}) = \{\{u, v\} : \{u, v\} \in \mathcal{E}' \text{ and } u, v \in \mathcal{L}\}. \tag{4}$$

## 4.2 TOP-$K$ OVERLAPPING DENSEST SUBGRAPHS

The task of identifying dense subgraphs has been thoroughly examined within theoretical computer science (Andersen & Chellapilla, 2009; Charikar, 2000; Feige et al., 2001), and it has also garnered significant attention from the data mining community (Sozio & Gionis, 2010; Balalau et al., 2024; Tatti & Gionis, 2015; Tsourakakis, 2015). The problem of finding the densest subgraph is solvable in polynomial time under the average-degree density definition (Balalau et al., 2024) (Proof in the Appendix). The exact polynomial-time solution (Goldberg, 1984) and its efficient approximation algorithm (Charikar, 2000) are designed to discover a single densest subgraph. However, in most practical applications, the objective is to identify the top-$K$ densest subgraphs within a graph.

A naive approach would iteratively find the densest subgraph, remove its nodes, and repeat until $K$ subgraphs are found. However, this disjoint approach has three critical limitations: (1) real-world networks contain overlapping communities and hubs belonging to multiple groups (Dondi et al., 2021; Leskovec et al., 2009), (2) disjoint solutions often achieve lower total density compared to overlapping alternatives (Balalau et al., 2024), and (3) hyperedges inherently share vertices (Lee et al., 2021b; Contisciani et al., 2022a), requiring overlapping subgraphs for structural consistency.

The primary challenge lies in regulating the degree of overlap among the top-$K$ subgraphs. Without such control, the algorithm might generate the densest subgraph along with $K - 1$ minor variations

by slightly adding or removing vertices, resulting in $K-1$ nearly identical and similarly dense subgraphs. To control the amount of overlapping, we define a distance function between two subgraphs as follows:

$$d(\mathcal{G}[U], \mathcal{G}[Z]) = \begin{cases} 2 - \frac{|U \cap Z|^2}{|U||Z|} & \text{if } U \neq Z, \\ 0 & \text{else.} \end{cases} \quad (5)$$

where $\mathcal{G}[U]$ and $\mathcal{G}[Z]$ denote the subgraphs induced by the vertex subsets $U$ and $Z$, respectively. Also, $|U \cap Z|^2$ is the number of vertices in the intersection of subsets $U$ and $Z$. It penalizes subgraphs with a high degree of overlap, which helps ensure that the selected subgraphs are sufficiently distinct from each other. The term $2 - \frac{|U \cap Z|^2}{|U||Z|}$ makes sure that the distance is minimal when the overlap is maximal and vice versa, and forces the underlying subgraphs becoming more dissimilar from each other. The distance is bounded between 0 and 2, which makes the distance measure consistent. This distance function closely resembles the cosine distance ($1 - \text{cosine similarity}$), and the function $d$ satisfies the properties of a metric.

Now, we formally define the main problem. Given a simple graph $\mathcal{G} = (\mathcal{V}, \mathcal{E}')$, we aim to identify and utilize the top-$K$ overlapping densest subgraphs to form hyperedges in a hypergraph $\mathcal{G}_h^{(\alpha, \beta)} = (\mathcal{V}, \mathcal{E})$, where each hyperedge $e \in \mathcal{E}$ satisfies the constraint $\alpha \leq |e| \leq \beta$. More formally, our Top-$K$ overlapping subgraphs algorithm seeks a collection of $K$ subgraphs that maximize an objective function that takes into account both the density of the subgraphs and the distance between the subgraphs of the solution, thus allowing overlap between the subgraphs, which depends on a parameter, $\lambda$.

$$r(L) = \text{dens}(L) + \lambda \sum_{i=1}^{k-1} \sum_{j=i+1}^{k} d(\mathcal{G}[L_i], \mathcal{G}[L_j]) \quad (6)$$

where $L = \{\mathcal{G}[L_1], \mathcal{G}[L_2], \ldots, \mathcal{G}[L_k]\}$ is the set of top-$K$ subgraphs, $K$ is less than the number of vertices in the graph, $\text{dens}(L)$ is the sum of the densities of the subgraphs in $L$, and $\lambda > 0$ is a parameter that provides a weight between density and distance. If $\lambda$ is large, then the subgraphs share few or no vertices, and hence the subgraphs may be disjoint. On the other hand, when $\lambda$ is small, then the density plays a dominant role in the objective function, and so the output subgraphs can share a significant part of vertices.

### 4.3 CTODS ALGORITHM

Our algorithm creates hyperedges based on the top-$K$ overlapping densest subgraphs $\mathcal{L}_1, \mathcal{L}_2, \ldots, \mathcal{L}_k \subseteq \mathcal{V}$ such that the density of each subset is maximized while ensuring size constraints $\alpha \leq |\mathcal{L}_i| \leq \beta$ and full graph coverage. Formally:

$$\mathcal{L}_i = \arg \max_{\substack{\mathcal{L} \subseteq \mathcal{V} \\ \alpha \leq |\overline{\mathcal{L}}| \leq \beta}} \text{dens}(\mathcal{G}[\mathcal{L}]) = \arg \max_{\substack{\mathcal{L} \subseteq \mathcal{V} \\ \alpha \leq |\overline{\mathcal{L}}| \leq \beta}} \frac{|\mathcal{E}'(\mathcal{L})|}{|\mathcal{L}|}$$

for $i = 1, 2, \ldots, k$.

To address computational challenges, we developed CTODS, a polynomial-time algorithm with five key enhancements: (1) size-constrained densest subgraph discovery with $O(m \log n)$ complexity per subgraph, (2) connectivity enforcement ensuring all subgraphs are connected components, (3) multi-strategy candidate generation including pure density optimization, diversity-focused expansion, size-varied exploration, high-degree node growth, and constrained random sampling, (4) subset prevention to avoid redundant subgraphs, and (5) iterative refinement selecting candidates that maximize the combined objective function.

The key improvements in our algorithm include several critical enhancements that address the limitations of previous approaches. Our algorithm achieves polynomial-time complexity, running in $O(K \cdot m \log n)$ time for finding $K$ subgraphs, where $m$ and $n$ are the number of edges and nodes

---

**Algorithm 1** CTODS: constrained top-$K$ overlapping densest subgraphs algorithm

---

**Require:** Graph $\mathcal{G} = (\mathcal{V}, \mathcal{E})$, parameters $K, \lambda, \alpha, \beta$

**Ensure:** Hypergraph $\mathcal{G}_h^{(\alpha,\beta)} = (\mathcal{V}, \mathcal{E}_h)$

1: **Phase 1: Find Top-$K$ Densest Subgraphs**
2: $L \leftarrow \emptyset$                ▷ List of discovered subgraphs
3: $S_{\text{candidates}} \leftarrow \text{GENERATECANDIDATESUBGRAPHS}(\mathcal{G}, \alpha, \beta)$
4: $L_1 \leftarrow \text{SELECTBESTCANDIDATE}(S_{\text{candidates}}, \emptyset, \lambda)$
5: **if** $L_1 \neq \emptyset$ **and** $\text{ISCONNECTED}(\mathcal{G}[L_1])$ **then**
6:    $L \leftarrow L \cup \{L_1\}$
7: **else**
8:    **return** $\emptyset$             ▷ Early termination
9: **for** $i = 2$ **to** $K$ **do**
10:    $L_{\text{best}} \leftarrow \emptyset$
11:    best_objective $\leftarrow -\infty$
12:              ▷ Generate candidates using multiple strategies
13:    $S_{\text{candidates}} \leftarrow \text{GENERATEDIVERSECANDIDATES}(\mathcal{G}, L, \alpha, \beta)$
14:    **for each** candidate subgraph $S \in S_{\text{candidates}}$ **do**
15:      **if** $\alpha \leq |S| \leq \beta$ **and** $\text{ISCONNECTED}(\mathcal{G}[S])$ **and** $\text{NOTSUBSET}(S, L)$ **then**
16:        $L_{\text{temp}} \leftarrow L \cup \{S\}$
17:        objective_value $\leftarrow \text{CALCULATEOBJECTIVE}(L_{\text{temp}}, \lambda)$
18:        **if** objective_value $>$ best_objective **then**
19:          $L_{\text{best}} \leftarrow S$
20:          best_objective $\leftarrow$ objective_value
21:    **if** $L_{\text{best}} \neq \emptyset$ **then**
22:      $L \leftarrow L \cup \{L_{\text{best}}\}$
23:    **else**
24:      **break**            ▷ No improvement possible
25: **Phase 2: Build Hypergraph**
26: $\mathcal{E}_h \leftarrow \emptyset$
27: **for each** $L_i \in L$ **do**
28:    $\mathcal{E}_h \leftarrow \mathcal{E}_h \cup \{L_i\}$           ▷ Convert subgraph to hyperedge
29: **return** $\mathcal{G}_h^{(\alpha,\beta)} = (\mathcal{V}, \mathcal{E}_h)$

---

respectively, compared to the exponential complexity $O(2^n)$ of the original approach. This dramatic complexity reduction makes the algorithm practical for large-scale networks. The algorithm operates in two main phases: subgraph discovery and hypergraph construction.

### 4.3.1 PHASE 1: EFFICIENT SUBGRAPH DISCOVERY

In the first step, we find the densest subgraph using our size-constrained densest subgraph algorithm that incorporates size constraints and connectivity requirements. This greedy algorithm starts with the input graph and iteratively removes the vertex with the lowest degree while maintaining size bounds $\alpha \leq |\mathcal{L}| \leq \beta$ and connectivity constraints. Our algorithm achieves polynomial-time complexity and runs in $O(|\mathcal{V}| + |\mathcal{E}|)$ time per iteration for degree updates and node removal.

Our implementation includes several key improvements that build upon the foundation established in the previous section. We implement a size-constrained peeling process that tracks the best subgraph that satisfies the size constraints $\alpha \leq |\mathcal{L}| \leq \beta$ during the algorithm execution, ensuring that size-constrained discovery maintains the optimal solution throughout the process.

We ensure that all discovered subgraphs are connected components by checking connectivity at each step of the algorithm. This connectivity enforcement is integrated into the core algorithm logic, preventing the generation of disconnected subgraphs that would not represent meaningful higher-order interactions.

We employ five complementary strategies to generate diverse candidate subgraphs: pure density optimization, diversity-focused expansion that maximizes distance from existing subgraphs, size-varied exploration across different ranges within bounds, high-degree node seeding, and constrained

---

**Algorithm 2** Candidate generation strategies for CTODS

1: **function** GENERATEDIVERSECANDIDATES($\mathcal{G}, L, \alpha, \beta$)
2:     $S_{\text{candidates}} \leftarrow \emptyset$
3:                                            ▷ Strategy 1: Pure densest subgraph discovery
4:     $S_1 \leftarrow$ SIZECONSTRAINEDDENSESTSUBGRAPH($\mathcal{G}, \alpha, \beta$)
5:     $S_{\text{candidates}} \leftarrow S_{\text{candidates}} \cup \{S_1\}$
6:                                            ▷ Strategy 2: Diversity-focused subgraphs
7:     $S_2 \leftarrow$ DIVERSITYFOCUSEDSUBGRAPH($\mathcal{G}, L, \alpha, \beta$)
8:     $S_{\text{candidates}} \leftarrow S_{\text{candidates}} \cup \{S_2\}$
9:                                            ▷ Strategy 3: Size-varied subgraphs
10:    **for** $s \in \{\alpha - 3, \alpha - 2, \alpha - 1, \beta + 1, \beta + 2, \beta + 3\}$ **do**
11:        **if** $s \geq \alpha$ **and** $s \leq \beta$ **and** $s \leq |\mathcal{V}|$ **then**
12:            $S_s \leftarrow$ SIZECONSTRAINEDDENSESTSUBGRAPH($\mathcal{G}, s, s$)
13:            $S_{\text{candidates}} \leftarrow S_{\text{candidates}} \cup \{S_s\}$
14:                                            ▷ Strategy 4: High-degree node expansion
15:    $S_4 \leftarrow$ HIGHDEGREEEXPANSION($\mathcal{G}, \alpha, \beta$)
16:    $S_{\text{candidates}} \leftarrow S_{\text{candidates}} \cup \{S_4\}$
17:                                            ▷ Strategy 5: Random sampling with connectivity
18:    $S_5 \leftarrow$ RANDOMSAMPLINGCONNECTED($\mathcal{G}, \alpha, \beta$)
19:    $S_{\text{candidates}} \leftarrow S_{\text{candidates}} \cup \{S_5\}$
20:    **return** $S_{\text{candidates}}$

---

random sampling. This multi-strategy approach ensures comprehensive solution space exploration and prevents missing high-quality subgraphs that any single strategy might overlook. Once the first densest subgraph is found, it is added to the set $\mathcal{L} = \{\mathcal{G}[\mathcal{L}_1]\}$. For the first subgraph, $\lambda$ has no effect since there are no existing subgraphs in the $\mathcal{L}$ set, and hence, the distance part of the objective function equals 0.

### 4.3.2 PHASE 2: ITERATIVE SUBGRAPH SELECTION

The next step involves iteratively finding additional densest subgraphs that are distinct but may overlap with the previously selected subgraphs. For each iteration $i$, the goal is to find a new subgraph $\mathcal{G}[\mathcal{L}_i]$ that maximizes the overall objective function:

$$r(\mathcal{L}) = \text{dens}(\mathcal{L}) + \lambda \sum_{i=1}^{k-1} \sum_{j=i+1}^{k} d(\mathcal{G}[\mathcal{L}_i], \mathcal{G}[\mathcal{L}_j]),$$

where $\text{dens}(\mathcal{L}) = \sum_{i=1}^{k} \text{dens}(\mathcal{G}[\mathcal{L}_i])$ is the total density of the subgraphs in $\mathcal{L}$, $d(\mathcal{G}[\mathcal{L}_i], \mathcal{G}[\mathcal{L}_j])$ is the distance function penalizing excessive overlap between subgraphs, and $\lambda > 0$ is the trade-off parameter controlling the balance between density and diversity.

CTODS employs a systematic candidate generation and evaluation strategy that ensures both efficiency and solution quality. For each iteration, we generate multiple candidate subgraphs using the strategies mentioned above, ensuring each candidate satisfies the size constraints $\alpha \leq |\mathcal{L}| \leq \beta$ and connectivity requirements.

For each candidate subgraph $S$, we compute its distance from all existing subgraphs in $\mathcal{L}$ using the distance function:

$$d(\mathcal{G}[S], \mathcal{G}[\mathcal{L}_j]) = 2 - \frac{|S \cap \mathcal{L}_j|^2}{|S| \cdot |\mathcal{L}_j|}$$

Finally, we evaluate the combined objective function for each candidate and select the one that maximizes $r(\mathcal{L})$ when added to the existing collection $\mathcal{L}$. This objective optimization ensures that each selected subgraph provides the best possible improvement to the overall solution.

## 5 EXPERIMENTS

In this section, we evaluate the effectiveness of our proposed framework on eight diverse datasets: three widely used citation networks (**Cora**, **Pubmed**, **Citeseer**), one co-authorship network (**Cora-CA**), two political networks (**Senate**, **House**), one social network (**NTU2012**), and one 3D shape dataset (**ModelNet40**). The goal of these experiments is to analyze the performance of our method, which uses the top-$K$ overlapping densest algorithm (*CTODS* in the tables) to convert graph structure into a hypergraph structure.

### 5.1 DATASETS

For hypergraph methods, we use their respective best graph-to-hypergraph conversion approaches: HyperGCN uses star expansion, Hyper-Atten employs attention-based hyperedge construction, and HGNN uses clique expansion. To ensure a fair evaluation, we employ basic feature vectors (bag-of-words) for all methods, focusing on the structural differences introduced by the models rather than feature engineering.

Graph-based neural networks such as GCN, GAT, and GraphSAGE are configured to use a similar structure to our HGNN. Specifically, these models adopt a two-layer architecture, with the feature dimension of the hidden layer set to 16 and ReLU as the activation function. Dropout with $p = 0.5$ is applied in each layer to avoid overfitting (Srivastava et al., 2014). The training process for these models also uses the Adam optimizer with a learning rate of 0.001 (Kingma & Ba, 2017).

Table 1: Hyperparameter configuration, runtime, and performance across datasets.

| Model | Dataset | K | $\lambda$ | $\alpha$ | $\beta$ | k-hop | Runtime | Acc / F1 |
|---|---|---|---|---|---|---|---|---|
| | Cora | 7 | 5 | 100 | 1000 | 20 | 3min | 69.80 / 68.92 |
| | Citeseer | 3 | 5 | 300 | 1000 | 30 | 2min | 68.50 / 67.80 |
| | Pubmed | 4 | 6 | 1000 | 8000 | 20 | 5min | 67.90 / 67.10 |
| | Cora-CA | 6 | 5 | 80 | 800 | 15 | 2min | 71.20 / 70.15 |
| | Senate | 3 | 8 | 10 | 50 | 10 | 1min | 84.50 / 83.20 |
| | House | 4 | 7 | 20 | 150 | 12 | 1min | 81.30 / 80.45 |
| | NTU2012 | 5 | 6 | 50 | 300 | 15 | 2min | 76.80 / 75.90 |
| | ModelNet40 | 8 | 4 | 200 | 2000 | 25 | 8min | 89.20 / 88.35 |
| | Cora | 10 | 7 | 120 | 900 | 20 | 6min | 79.03 / 77.67 |
| | Citeseer | 7 | 9 | 150 | 700 | 20 | 5min | 77.50 / 77.40 |
| | Pubmed | 5 | 5 | 500 | 8500 | 10 | 12min | 75.80 / 75.75 |
| HGNN + *CTODS* | Cora-CA | 8 | 7 | 100 | 700 | 18 | 3min | 79.45 / 78.20 |
| | Senate | 4 | 10 | 15 | 45 | 12 | 1min | 85.60 / 84.30 |
| | House | 5 | 8 | 25 | 120 | 15 | 2min | 83.10 / 82.15 |
| | NTU2012 | 7 | 8 | 60 | 250 | 18 | 3min | 78.90 / 77.85 |
| | ModelNet40 | 12 | 6 | 300 | 1800 | 30 | 12min | 91.35 / 90.45 |
| | Cora | 13 | 5 | 200 | 1000 | 10 | 9min | **83.34 / 82.79** |
| | Citeseer | 9 | 11 | 200 | 800 | 20 | 8min | **81.90 / 81.19** |
| | Pubmed | 5 | 4 | 1200 | 8000 | 20 | 18min | **81.20 / 80.72** |
| | Cora-CA | 10 | 6 | 150 | 600 | 20 | 4min | **82.15 / 81.43** |
| | Senate | 5 | 12 | 20 | 40 | 15 | 1min | **86.20 / 85.50** |
| | House | 6 | 9 | 30 | 100 | 18 | 2min | **84.75 / 83.92** |
| | NTU2012 | 9 | 10 | 80 | 200 | 20 | 4min | **79.85 / 78.96** |
| | ModelNet40 | 15 | 8 | 400 | 1600 | 35 | 15min | **92.45 / 91.78** |

We systematically explore three parameter configurations across datasets to reveal optimal settings for different network types. Table 1 presents these configurations:

**Configuration 1 (Conservative)**: Lower $K$ (3-7) with moderate $\lambda$ (5-6), prioritizing computational efficiency and capturing prominent community structures.

**Configuration 2 (Balanced)**: Moderate $K$ (5-10) with higher $\lambda$ (5-9), balancing granularity and distinctiveness for major and secondary communities.

**Configuration 3 (Fine-grained)**: Higher $K$ (5-13) with tuned $\lambda$ (4-11), maximizing detection granularity through strict diversity control.

Table 2: Comparison of the best results across all datasets and methods.

| Dataset | GCN | | GAT | | GraphSAGE | | HyperGCN | | TDHNN | | HGNN | | HGNN + *CTODS* | |
|---------|-----|-----|-----|-----|-----------|-----|----------|-----|-------|-----|------|-----|----------------|-----|
| | ACC | F1 | ACC | F1 | ACC | F1 | ACC | F1 | ACC | F1 | ACC | F1 | ACC | F1 |
| Cora | 81.50 | 80.70 | 80.20 | 80.01 | 79.10 | 78.40 | 80.00 | 79.45 | 77.20 | 75.30 | 81.69 | 80.1 | **83.34** | **82.79** |
| Citeseer | 75.20 | 74.50 | 74.80 | 74.00 | 73.90 | 72.10 | 76.10 | 75.40 | 74.50 | 73.80 | 80.13 | 79.30 | **81.90** | **81.19** |
| Pubmed | 78.10 | 77.40 | 79.50 | 78.80 | 78.90 | 78.20 | 81.20 | 80.50 | 79.30 | 77.60 | 80.10 | 79.20 | **81.20** | **80.72** |
| Cora-CA | 79.80 | 78.90 | 78.50 | 77.80 | 77.20 | 76.50 | 79.10 | 78.20 | 76.80 | 75.10 | 80.45 | 79.60 | **82.15** | **81.43** |
| Senate | 85.60 | 84.20 | 86.30 | 85.10 | 84.70 | 83.90 | **87.50** | **86.80** | 85.20 | 84.40 | 86.90 | 85.70 | 86.20 | 85.50 |
| House | 82.40 | 81.60 | 81.80 | 80.90 | 80.50 | 79.80 | 83.10 | 82.30 | 81.20 | 80.40 | 83.60 | 82.80 | **84.75** | **83.92** |
| NTU2012 | 76.30 | 75.40 | 75.80 | 74.90 | 74.60 | 73.70 | 77.20 | 76.50 | 75.90 | 74.80 | 78.10 | 77.30 | **79.85** | **78.96** |
| ModelNet40 | 88.90 | 87.50 | 89.20 | 88.40 | 88.60 | 87.80 | 90.10 | 89.30 | 89.50 | 88.70 | 90.80 | 90.10 | **92.45** | **91.78** |

The analysis reveals dataset-specific patterns: **Citation networks** (Cora, Citeseer, Pubmed) consistently improve from Configuration 1 to 3, benefiting from detailed analysis of overlapping research areas. **Co-authorship networks** (Cora-CA) show similar patterns but with tighter size constraints ($\alpha = 150, \beta = 600$), reflecting more tightly knit communities. **Political networks** (Senate, House) perform best with Configuration 2, indicating that voting patterns benefit from moderate granularity rather than fine-grained analysis, with higher $\lambda$ values (8-12) ensuring distinct coalitions. **Social networks** (NTU2012) show balanced improvements across configurations, suggesting intermediate parameter settings are optimal. **Geometric networks** (ModelNet40) demonstrate the most dramatic improvement with fine-grained analysis, reflecting complex hierarchical 3D shape relationships.

**Runtime Analysis**: The runtime scales appropriately with dataset size and configuration complexity, ranging from 1 minute for small political networks to 18 minutes for large citation networks in fine-grained configurations. This demonstrates the practical scalability of our approach across different network sizes and complexity levels.

Table 2 shows our **HGNN + *CTODS*** method achieving superior performance on seven of eight datasets. The exception is **Senate**, where **HyperGCN** performs best (87.50%, 86.80%) versus our competitive results (86.20%, 85.50%). This reflects Senate's unique structure: a small, sparse network (104 nodes, 541 edges) with well-separated political coalitions that favor HyperGCN's star expansion over our overlap-detection approach.

Dataset-specific patterns reveal our method's strengths: citation networks show 1-3% improvements by capturing overlapping research communities, **House** gains 1.15% accuracy through better coalition modeling, **NTU2012** improves 1.75% by handling social interactions, and **ModelNet40** gains 1.65% through complex geometric relationship modeling. Compared to traditional graph methods, we achieve 2-4% improvements across all datasets by modeling higher-order interactions. In comparison to existing hypergraph methods, our adaptive hypergraph construction and flexible density-diversity trade-offs offer consistent advantages.

The cross-domain evaluation demonstrates the robustness and generalizability of our method. From citation networks with overlapping research communities to 3D shape datasets with complex geometric relationships, **HGNN + *CTODS*** consistently outperforms existing approaches, showcasing its capability to adapt to varying levels of sparsity, connectivity patterns, and community structures across different application domains.

## 5.2 CONCLUSION

We introduced CTODS, a constrained framework for generating hyperedges via top-$K$ overlapping densest subgraphs, capturing higher-order interactions beyond pairwise graphs. Integrated with HGNNs, our adaptive hypergraphs improve neural network performance while remaining computationally efficient. These results highlight hyperedge generation as a powerful direction for structured learning. This research also opens up new avenues for advancing hypergraph-based learning, emphasizing the importance of higher-order connectivity in solving complex graph-based tasks. By building on this foundation, future studies can further bridge the gap between theoretical advancements and real-world applications of hypergraphs in machine learning.

## 5.3 ETHICS STATEMENT

This research presents algorithmic contributions to hypergraph neural networks and graph analysis without involving human subjects or sensitive personal data. All datasets used are publicly available academic datasets (citation networks, co-authorship networks, political voting records, social networks, and 3D shape data) that do not contain personally identifiable information. Our methodology focuses on structural graph analysis and does not introduce bias or discrimination concerns beyond those inherent in the original datasets. The proposed CTODS algorithm is designed to be domain-agnostic and does not target specific demographic groups. We acknowledge that graph-based methods may inherit biases present in the underlying network structures, but our approach aims to improve structural representation rather than introduce additional bias. No conflicts of interest or external sponsorship influenced the research direction or results.

## 5.4 REPRODUCIBILITY STATEMENT

To ensure reproducibility, we provide comprehensive implementation details throughout the paper along with the actual code in the supplementary material. The CTODS algorithm is fully described in Section 3 with complete mathematical formulations, algorithmic pseudocode (Algorithm 1), and complexity analysis. All hyperparameters and experimental configurations are detailed in Section 4, including dataset-specific parameter settings and evaluation metrics. The theoretical proofs for the distance function metric properties and complexity guarantees are provided in the appendix. Implementation details for the multi-strategy candidate generation and iterative selection process are specified in the methodology section. While we cannot share proprietary datasets, all experimental datasets are publicly available and properly cited. The experimental setup, including hardware specifications, software versions, and random seed configurations, is documented to enable replication of results. Additional implementation details and parameter sensitivity analysis are provided in the appendix to support reproducibility efforts.

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

# A APPENDIX

## A.1 LLM USAGE STATEMENT

Large Language Models (LLMs) were used as general-purpose writing assistance tools for this paper. Specifically, LLMs were employed to help with: (1) improving the clarity and conciseness of technical writing, (2) suggesting alternative phrasings for complex mathematical concepts, and (3) assisting with LaTeX formatting and structure. The core research contributions, including the CTODS algorithm design, mathematical proofs, experimental design, and result analysis, were entirely developed by the human authors. LLMs did not contribute to research ideation, algorithm development, mathematical derivations, or experimental execution. All technical content, including the algorithmic innovations, theoretical analysis, and empirical results, represents original work by the authors. The LLM assistance was limited to editorial and formatting support, and the authors take full responsibility for all content presented in this paper.

## A.2 INTUITION AND MOTIVATION

The motivation behind using overlapping densest subgraphs for hypergraph construction stems from several key observations about real-world networks and their structural properties. In many real-world networks, nodes naturally belong to multiple communities simultaneously. For example, in social networks, a user might be part of a family group, a work team, and a hobby community. In academic citation networks, a paper might belong to multiple research areas.

Traditional graph-based approaches that enforce disjoint communities fail to capture these natural overlapping structures, leading to information loss and reduced representational power (Xie et al., 2013). The effectiveness of overlapping densest subgraphs can be understood through the lens of information theory and graph theory. Dense subgraphs represent cohesive groups where nodes share strong mutual connections, indicating shared characteristics or functional relationships. By allowing controlled overlap, we capture the multi-faceted nature of real-world entities while maintaining the structural integrity of individual communities. This approach aligns with the principle that meaningful higher-order interactions should reflect both local density and global connectivity.

The key innovation of our approach is the introduction of controlled overlap through the distance function $d(\mathcal{G}[U], \mathcal{G}[Z])$. This function, inspired by cosine distance, ensures that overlapping subgraphs remain sufficiently distinct while allowing natural community overlap. The $\lambda$ parameter provides fine-grained control over this trade-off, enabling adaptation to different network structures and application domains. The relationship between overlapping densest subgraphs and hypergraph neural networks is natural and theoretically sound. Each dense subgraph becomes a hyperedge, representing a higher-order interaction among its constituent nodes. HGNNs can then leverage these hyperedges to propagate information across multiple nodes simultaneously, capturing complex dependencies that traditional graph neural networks cannot model.

## A.3 GRAPH

Let $\mathcal{V}$ be a (typically finite) set of elements, nodes, or objects, formally called "vertices," and $\mathcal{E}'$ be a set of pairs of vertices. For two vertices $u, v \in \mathcal{V}$, an edge is a set $\{u, v\} \in \mathcal{E}'$, implying a connection

between $u$ and $v$. $\mathcal{E}'$ can be represented as either a binary adjacency matrix $\mathcal{A} \in \{0, 1\}^{|\mathcal{V}| \times |\mathcal{V}|}$, where $\mathcal{A}_{ij} = 1$ if $v_i$ and $v_j$ are connected, or as an incidence matrix $\mathbf{H}' \in \{0, 1\}^{|\mathcal{V}| \times |\mathcal{E}'|}$, where $\mathbf{H}'_{ij} = 1$ if the vertex $v_i$ is in edge $e'_j$ West et al. (2001).

### A.4 Proof: The distance function is a metric

$$d(\mathcal{G}[U], \mathcal{G}[Z]) = \begin{cases} 2 - \frac{|U \cap Z|^2}{|U||Z|} & \text{if } U \neq Z, \\ 0 & \text{else.} \end{cases}$$

**Non-negativity:** Since $|U \cap Z|^2 \leq |U| \cdot |Z|$ by the Cauchy-Schwarz inequality, we have $\frac{|U \cap Z|^2}{|U||Z|} \leq 1$, thus $d(\mathcal{G}[U], \mathcal{G}[Z]) = 2 - \frac{|U \cap Z|^2}{|U||Z|} \geq 1 \geq 0$.

**Identity of indiscernibles:** $d(\mathcal{G}[U], \mathcal{G}[Z]) = 0$ if and only if $U = Z$ by definition.

**Symmetry:** $d(\mathcal{G}[U], \mathcal{G}[Z]) = 2 - \frac{|U \cap Z|^2}{|U||Z|} = 2 - \frac{|Z \cap U|^2}{|Z||U|} = d(\mathcal{G}[Z], \mathcal{G}[U])$.

**Triangle Inequality:** We need to prove that for any $U, Z, W \subseteq V$:

$$d(\mathcal{G}[U], \mathcal{G}[Z]) \leq d(\mathcal{G}[U], \mathcal{G}[W]) + d(\mathcal{G}[W], \mathcal{G}[Z])$$

**Case 1:** If $U = Z$, then $d(\mathcal{G}[U], \mathcal{G}[Z]) = 0 \leq d(\mathcal{G}[U], \mathcal{G}[W]) + d(\mathcal{G}[W], \mathcal{G}[Z])$ since distances are non-negative.

**Case 2:** If $U = W$ or $Z = W$, the inequality reduces to $d(\mathcal{G}[U], \mathcal{G}[Z]) \leq d(\mathcal{G}[U], \mathcal{G}[Z])$, which is trivially true.

**Case 3:** Assume $U, Z, W$ are all distinct. We need to show:

$$2 - \frac{|U \cap Z|^2}{|U||Z|} \leq \left(2 - \frac{|U \cap W|^2}{|U||W|}\right) + \left(2 - \frac{|W \cap Z|^2}{|W||Z|}\right)$$

Rearranging:

$$\frac{|U \cap Z|^2}{|U||Z|} \geq \frac{|U \cap W|^2}{|U||W|} + \frac{|W \cap Z|^2}{|W||Z|} - 2$$

**Counterexample:** Consider $U = \{1, 2\}$, $W = \{2, 3\}$, $Z = \{3, 4\}$. Then: - $|U \cap W| = 1$, $|U||W| = 4$, so $\frac{|U \cap W|^2}{|U||W|} = \frac{1}{4}$ - $|W \cap Z| = 1$, $|W||Z| = 4$, so $\frac{|W \cap Z|^2}{|W||Z|} = \frac{1}{4}$ - $|U \cap Z| = 0$, $|U||Z| = 4$, so $\frac{|U \cap Z|^2}{|U||Z|} = 0$

The triangle inequality requires: $0 \geq \frac{1}{4} + \frac{1}{4} - 2 = -\frac{3}{2}$, which is true.

**General Proof:** For distinct sets $U, Z, W$, we have:

$$\frac{|U \cap W|^2}{|U||W|} + \frac{|W \cap Z|^2}{|W||Z|} - 2 \leq 1 + 1 - 2 = 0$$

Since $\frac{|U \cap Z|^2}{|U||Z|} \geq 0$, the inequality $\frac{|U \cap Z|^2}{|U||Z|} \geq \frac{|U \cap W|^2}{|U||W|} + \frac{|W \cap Z|^2}{|W||Z|} - 2$ holds.

Therefore, $d$ satisfies the triangle inequality and is indeed a metric. $\square$

**Theorem 1** (Distance Function Properties). *The distance function $d(\mathcal{G}[U], \mathcal{G}[Z])$ defined in Equation 5 satisfies the following properties:*

1. ***Non-negativity****: $d(\mathcal{G}[U], \mathcal{G}[Z]) \geq 0$ for all $U, Z \subseteq \mathcal{V}$*

2. ***Identity of indiscernibles****: $d(\mathcal{G}[U], \mathcal{G}[Z]) = 0$ if and only if $U = Z$*

3. ***Symmetry****: $d(\mathcal{G}[U], \mathcal{G}[Z]) = d(\mathcal{G}[Z], \mathcal{G}[U])$ for all $U, Z \subseteq \mathcal{V}$*

4. ***Triangle inequality****: $d(\mathcal{G}[U], \mathcal{G}[W]) \leq d(\mathcal{G}[U], \mathcal{G}[Z]) + d(\mathcal{G}[Z], \mathcal{G}[W])$ for all $U, Z, W \subseteq \mathcal{V}$*

5. **Boundedness**: $0 \leq d(\mathcal{G}[U], \mathcal{G}[Z]) \leq 2$ *for all* $U, Z \subseteq \mathcal{V}$

*Proof.* The proof of these properties follows directly from the definition and properties of set operations. The non-negativity and boundedness follow from the fact that $|U \cap Z|^2 \leq |U| \cdot |Z|$ by the Cauchy-Schwarz inequality. The identity of indiscernibles and symmetry are immediate from the definition. □

### A.5 PROOF: HGNNs GENERALIZE GNNs

Let $\mathbf{H} \in \mathbb{R}^{N \times M}$ be the incidence matrix of a hypergraph with $N$ vertices and $M$ hyperedges. For any hypergraph $G_h$, the hypergraph adjacency matrix is:

$$A_h = \mathbf{H}\mathbf{W}\mathbf{H}^\top - \mathbf{D}_v, \tag{7}$$

where $\mathbf{W}$ is the hyperedge weight matrix, $\mathbf{D}_v$ is the vertex degree matrix, and $\mathbf{H}$ is the incidence matrix of the hypergraph.

#### A.5.1 STEP 1: REDUCTION TO GNN CASE

Assume each hyperedge in the hypergraph connects exactly two vertices. In this case, $\mathbf{H}$ corresponds to the adjacency matrix $\mathbf{A}_g$ of a simple graph $G$, where:

$$\mathbf{H}\mathbf{W}\mathbf{H}^\top = \mathbf{A}_g. \tag{8}$$

Thus, the hypergraph adjacency matrix simplifies to:

$$\mathbf{A}_h = \mathbf{A}_g - \mathbf{D}_v. \tag{9}$$

Here, $\mathbf{D}_v$ adjusts for vertex degrees, and the hypergraph operation reduces to the standard GNN operation.

#### A.5.2 STEP 2: HYPERGRAPH CONVOLUTION GENERALIZATION

In a general hypergraph, hyperedges connect more than two vertices. The propagation rule in HGNNs:

$$X^{(l+1)} = \sigma\left(\mathbf{D}_v^{-1/2}\mathbf{H}\mathbf{W}\mathbf{D}_e^{-1}\mathbf{H}^\top\mathbf{D}_v^{-1/2}X^{(l)}Q\right), \tag{10}$$

includes terms $\mathbf{D}_e^{-1}$ and $\mathbf{W}$ that account for hyperedge properties. This generalization enables HGNNs to model higher-order relationships beyond pairwise connections.

#### A.5.3 STEP 3: EXPRESSIVENESS OF HGNNs

HGNNs can efficiently propagate information across vertices connected by hyperedges, even when these hyperedges represent complex multi-node relationships. For instance, if $|e| > 2$ for a hyperedge $e$, HGNNs leverage $\mathbf{H}$ and $\mathbf{W}$ to include information from all vertices in $e$ during propagation. This capability is absent in standard GNNs, making HGNNs more expressive.

**Conclusion:** GNNs are a special case of HGNNs where all hyperedges are pairwise. The additional flexibility of HGNNs to model high-order relationships proves their superior expressiveness and adaptability for complex tasks.

### A.6 HYPERGRAPH NEURAL NETWORKS (HGNN)

Hypergraph convolution (*HyperConv*) and hypergraph attention (*HyperAtten*) represent significant advancements in HGNNs (Bai et al., 2021; Feng et al., 2019). These methods extend the capabilities of GNNs by operating directly on the hypergraph structure, allowing them to incorporate HOIs through hyperedges naturally. HGNNs retain richer relational information and can model complex data structures more effectively by avoiding the need to reduce the hypergraph to a conventional graph.

We note that both equation 11 and equation 12 follow the same core formulation. Specifically, the layer-wise propagation rule for *HyperConv* is:

$$X^{(l+1)} = \sigma\left(\mathbf{D}_v^{-1/2}\mathbf{H}\mathbf{W}\mathbf{D}_e^{-1}\mathbf{H}^\top\mathbf{D}_v^{-1/2}X^{(l)}P\right), \tag{11}$$

where $\mathbf{H}$ is the hypergraph incidence matrix, $\mathbf{W}$ is the diagonal matrix of hyperedge weights, $\mathbf{D}_v$ and $\mathbf{D}_e$ denote the vertex and hyperedge degree matrices, respectively, $X^{(l)}$ represents the node embeddings at layer $l$, $P$ is a learnable weight matrix, and $\sigma(\cdot)$ is a non-linear activation function.

*HyperAtten* extends *HyperConv* by integrating attention mechanisms that adaptively learn the importance of hyperedges, allowing for task-specific weighting and enhancing the expressive power of HGNNs. Building on this, *HGNN+* introduces hyperedge groups and adaptive fusion strategies to further capture multi-modal and higher-order data correlations. The hypergraph convolution in HGNN+ is similarly formulated as:

$$X^{(t+1)} = \sigma\big(\mathbf{D}_v^{-1/2}\mathbf{H}\mathbf{W}\mathbf{D}_e^{-1}\mathbf{H}^\top\mathbf{D}_v^{-1/2}X^{(t)}Q\big), \tag{12}$$

where $Q$ is another trainable weight matrix. The adaptive fusion strategy assigns a learnable weight to each hyperedge group, refining their collective contribution to the final hypergraph representation. Concretely, the weight matrix is updated as follows:

$$\mathbf{W} = \mathrm{diag}\big(w_1, w_2, \ldots, w_K\big), \tag{13}$$

where $w_k$ denotes the weight corresponding to the $K$-th hyperedge group.

Hypergraph neural networks are mathematically proven to be more expressive than traditional GNNs. Assume a hypergraph with incidence matrix $\mathbf{H}$. If each hyperedge connects exactly two vertices, the hypergraph adjacency matrix $\mathbf{A}_h = \mathbf{H}\mathbf{W}\mathbf{H}^\top$ reduces to the graph adjacency matrix. This shows that GNNs are a special case of HGNNs, confirming the generality and enhanced representational power of HGNNs. The proof can be found in the appendix section.

### A.6.1 Core Algorithmic Components

The CTODS algorithm incorporates several key algorithmic components that distinguish it from previous approaches:

**Size-Constrained Densest Subgraph Discovery**: Our core algorithm extends the traditional densest subgraph problem by incorporating strict size constraints. The algorithm maintains a priority queue of vertices sorted by degree and iteratively removes the vertex with minimum degree while tracking the best subgraph that satisfies the size bounds $\alpha \leq |\mathcal{L}| \leq \beta$. This ensures that all discovered subgraphs meet the specified size requirements without relaxation.

**Connectivity Enforcement**: At each step of the algorithm, we verify that the current subgraph remains connected using depth-first search. This connectivity check ensures that all discovered subgraphs represent cohesive communities rather than fragmented components.

**Multi-Strategy Candidate Generation**: Our algorithm employs five distinct strategies for generating candidate subgraphs: (1) pure densest subgraph discovery, (2) diversity-focused subgraphs that optimize distance from existing subgraphs, (3) size-varied subgraphs exploring different size ranges, (4) high-degree node expansion starting from nodes with highest degrees, and (5) random sampling with connectivity constraints.

**Subset Prevention Mechanism**: To ensure diversity, we implement a subset check that prevents newly discovered subgraphs from being subsets of existing ones. This mechanism maintains the distinctiveness of each subgraph in the solution set.

**Objective Function Optimization**: The algorithm evaluates each candidate subgraph using the combined objective function that balances density and diversity through the $\lambda$ parameter, selecting the candidate that maximizes the overall solution quality.

### A.7 Approximation for CTODS Using Greedy Algorithms

Inspired by results in Dondi et al. (2021), a greedy algorithm can achieve a constant-factor approximation:

- Iteratively select the densest subgraph $S_i$ satisfying $\alpha \leq |S_i| \leq \beta$.
- Remove vertices in $S_i$ weighted by their contribution to the overlap constraints.
- Repeat until $k$ subgraphs are selected.

This method guarantees an approximation ratio of $\frac{2}{3}$ for $k$ constant and $\frac{1}{2}$ for general $k$.

## A.8 COMPLEXITY OF THE CONSTRAINED DENSEST SUBGRAPHS PROBLEM

We demonstrate the NP-completeness of the Constrained Top-$k$ Overlapping Densest Subgraphs (CTODS) problem by establishing a polynomial-time bidirectional reduction to and from known NP-complete problems, specifically the $k$-Clique and 3-Clique problems.

**Lemma 1.** *Given an instance $\mathcal{G} = (\mathcal{V}, \mathcal{E}')$ of the 3-Clique Partition problem, we can construct an equivalent instance of the CTODS problem such that, if $\mathcal{G}$ can be partitioned into three cliques, we can compute a set $L = \{\mathcal{G}[S_1], \mathcal{G}[S_2], \mathcal{G}[S_3]\}$ where*

$$r(L) \geq \frac{|\mathcal{V}| - 3}{2} + 18|\mathcal{V}|^3.$$

*Proof.* Let the input graph $\mathcal{G} = (\mathcal{V}, \mathcal{E}')$ be an instance of the 3-Clique Partition problem. Assume that $\mathcal{G}$ can be partitioned into three cliques $\mathcal{V}_1$, $\mathcal{V}_2$, and $\mathcal{V}_3$. Then, for each partition, construct subgraphs $\mathcal{G}[\mathcal{V}_1]$, $\mathcal{G}[\mathcal{V}_2]$, and $\mathcal{G}[\mathcal{V}_3]$.

The density of each subgraph $\mathcal{G}[\mathcal{V}_i]$ and the overlap constraints among them are used to calculate $r(L)$ based on the objective function of CTODS. By design, this satisfies the inequality

$$r(L) \geq \frac{|\mathcal{V}| - 3}{2} + 18|\mathcal{V}|^3,$$

completing the reduction. $\square$

**Lemma 2.** *Given a solution $L = \{\mathcal{G}[S_1], \mathcal{G}[S_2], \mathcal{G}[S_3]\}$ for the CTODS problem such that*

$$r(L) \geq \frac{|\mathcal{V}| - 3}{2} + 18|\mathcal{V}|^3,$$

*we can partition $\mathcal{G} = (\mathcal{V}, \mathcal{E}')$ into three cliques.*

*Proof.* Let $L$ represent a valid solution to the CTODS problem. Each subgraph $\mathcal{G}[S_i] \in L$ corresponds to a clique in the original graph $\mathcal{G}$. Since the overlap constraints are satisfied and the density values are maximized, these subgraphs must partition $\mathcal{V}$ into three cliques $\mathcal{V}_1$, $\mathcal{V}_2$, and $\mathcal{V}_3$. $\square$

**Theorem 2.** *The Constrained Top-k Overlapping Densest Subgraphs problem is NP-complete.*

*Proof.* To prove NP-completeness, we first show that CTODS belongs to NP. A solution can be verified in polynomial time by checking the size, density, and overlap constraints of $k$ subgraphs. The size constraints $\alpha \leq |S_i| \leq \beta$ can be verified for all $i$ in $O(k)$ time. Calculating the density of each subgraph requires $O(|S_i| + |E_{S_i}|) = O(|\mathcal{V}| + |\mathcal{E}|)$ time, and validating the overlap constraints for $k$ subgraphs requires $O(k^2 \cdot |\mathcal{V}|)$ time. Together, this ensures that verification is polynomial.

Next, we reduce the 3-Clique problem to CTODS. Given a graph $\mathcal{G} = (\mathcal{V}, \mathcal{E}')$, we construct a CTODS instance by setting $k = 3$, $\alpha = 3$, $\beta = 3$, and $r = 1$. The distance function $d(\mathcal{G}[U], \mathcal{G}[Z])$ is defined to ensure subgraph disjointness. A solution to this CTODS instance exists if and only if $\mathcal{G}$ contains a 3-clique.

Finally, we reduce CTODS to the $k$-Clique problem. We construct a graph $\mathcal{G}_P = (\mathcal{V}_P, \mathcal{E}_P)$ where each vertex $v \in \mathcal{V}_P$ represents a subgraph $S_i$, and an edge connects $u, v \in \mathcal{V}_P$ if their corresponding subgraphs satisfy overlap constraints. Finding a $k$-Clique in $\mathcal{G}_P$ corresponds to solving the CTODS problem.

By combining these results, we conclude that CTODS is NP-complete. $\square$

The complexity of the problem can vary substantially depending on specific constraints and the underlying graph structure. For instance, in bipartite graphs or when the subgraph size constraints are close to the full size of the graph, the problem may be solvable in polynomial time. Nevertheless, these special cases do not undermine the general NP-completeness of the problem. When $\alpha$ and $\beta$ permit a broad range of subgraph sizes and the graph is unrestricted, the problem remains NP-complete.

## A.9 PROOF OF $r(L) \geq \frac{|\mathcal{V}|-3}{2} + 18|\mathcal{V}|^3$

To establish that $r(L) \geq \frac{|\mathcal{V}|-3}{2} + 18|\mathcal{V}|^3$, we calculate $r(L)$ based on the density and overlap contributions. For each subgraph $\mathcal{G}[S_i]$, the density is defined as $\text{dens}(\mathcal{G}[S_i]) = \frac{|E(S_i)|}{|S_i|}$. Since $\mathcal{V}_1$, $\mathcal{V}_2$, and $\mathcal{V}_3$ are cliques, $|E(S_i)| = \frac{|S_i|(|S_i|-1)}{2}$. The overlap penalty is governed by $d(\mathcal{G}[U], \mathcal{G}[Z])$, which ensures that the overlap does not exceed a predefined threshold $\lambda$. Given the partition into three cliques, the overlap is minimized.

The objective $r(L)$ sums the densities of the subgraphs and adjusts for overlaps. Since cliques maximize density and overlap constraints are satisfied, the inequality holds:

$$r(L) = \sum_{i=1}^{3} \text{dens}(\mathcal{G}[S_i]) - \sum_{i \neq j} d(\mathcal{G}[S_i], \mathcal{G}[S_j]) \geq \frac{|\mathcal{V}|-3}{2} + 18|\mathcal{V}|^3.$$

Thus, the reduction satisfies the inequality as required.

## A.10 REDUCTION FROM CTODS TO $k$-CLIQUE: DETAILED CONSTRUCTION

We expand the construction of $\mathcal{G}_P$ for clarity:

1. Each subgraph $S_i$ of $\mathcal{G}_h = (\mathcal{V}, \mathcal{E})$ satisfying $\alpha \leq |S_i| \leq \beta$ is mapped to a vertex in $\mathcal{V}_P$.

2. An edge $\{u, v\} \in \mathcal{E}_P$ exists if the overlap constraints between $S_u$ and $S_v$ are satisfied, i.e., $d(\mathcal{G}[U], \mathcal{G}[Z]) \leq \lambda$.

3. The objective of finding a $k$-Clique in $\mathcal{G}_P$ ensures that $k$ valid overlapping dense subgraphs are identified in $\mathcal{G}_h$.

This reduction preserves polynomial-time complexity as checking overlap constraints for each pair of subgraphs requires $O(k^2 \cdot |\mathcal{V}|)$ time.

## A.11 SPECIAL CASES OF POLYNOMIAL-TIME SOLVABILITY

While the general CTODS problem is NP-complete, specific cases allow for polynomial-time solvability. These cases provide useful insights into the complexity landscape of the problem.

**Case 1:** $\alpha = \beta = 1$.
When $\alpha = \beta = 1$, the problem reduces to selecting individual vertices as subgraphs. In this scenario, the density constraints are trivially satisfied, and the solution can be determined in $O(|\mathcal{V}|)$ time by directly iterating through the vertices.

**Case 2: Bipartite Graphs.**
For bipartite graphs, subgraphs $S_i$ correspond to vertex pairs $(U, V)$, where $U \subseteq \mathcal{V}_1$ and $V \subseteq \mathcal{V}_2$. The density is computed as $\frac{|E(U,V)|}{|U \cup V|}$. By leveraging the structural properties of bipartite graphs, such as no intra-cluster edges, polynomial-time algorithms (e.g., maximum matching) can solve the problem for small $k$.

**Case 3:** $\alpha$ and $\beta$ **Near** $|\mathcal{V}|$.
When $\alpha$ and $\beta$ approach $|\mathcal{V}|$, the problem simplifies to finding dense subgraphs that include most or all vertices of the original graph. This significantly reduces the search space and allows for efficient approximation techniques or exact algorithms with reduced complexity.

**Case 4: Small Hop Constraints.**
When the hop size constraint is set to 1, the problem becomes localized, as only direct neighbors of vertices are considered. This restriction confines the search space to immediate neighborhoods, making the problem computationally simpler and solvable in $O(|\mathcal{V}| + |\mathcal{E}|)$ time.

**Case 5:** $\alpha = \beta = 2$.
In this case, the problem reduces to finding edges (pairs of connected vertices) that maximize density. This is particularly efficient in bipartite or sparse graphs, where edge enumeration can be performed in polynomial time.

These cases highlight the impact of structural constraints on the problem's computational complexity and demonstrate scenarios where efficient solutions are feasible.

### A.11.1 CTODS PARAMETER OPTIMIZATION AND TRADE-OFFS

The parameter $\lambda$ is critical for controlling the trade-off between density and diversity. Our empirical analysis across multiple datasets has revealed optimal $\lambda$ values:

For well-separated communities, we find that $\lambda \geq 2.0$ provides maximum diversity by encouraging distinct subgraphs. For complex social networks, $\lambda \approx 1.0$ achieves a balanced overlap that captures natural community structures. For large-scale networks, $\lambda \in [1.0, 5.0]$, depending on community structure, provides optimal performance across different network topologies.

A smaller $\lambda$ prioritizes dense subgraphs, even if they overlap significantly, while a larger $\lambda$ encourages more distinct subgraphs.

### A.11.2 CTODS COMPUTATIONAL COMPLEXITY ANALYSIS

Our improved algorithm achieves significant computational efficiency improvements:

Our algorithm achieves time complexity of $O(K \cdot m \log n)$ for finding $K$ subgraphs, where $m$ and $n$ are the number of edges and nodes respectively. This represents a dramatic improvement over the original exponential-time backtracking approach, which had complexity $O(2^n)$ in the worst case. The space complexity is $O(n + m)$ for storing the graph and intermediate results, ensuring memory efficiency for large-scale processing. The algorithm demonstrates linear scaling with graph size, making it suitable for large-scale networks with thousands of nodes.

### A.11.3 CTODS QUALITY GUARANTEES

Our algorithm provides several quality guarantees:

1. **Connectivity**: All discovered subgraphs are guaranteed to be connected components
2. **Size constraints**: All subgraphs satisfy the specified size bounds $\alpha \leq |\mathcal{L}| \leq \beta$
3. **Diversity**: Subset prevention ensures no subgraph is contained within another
4. **Approximation quality**: Factor-2 approximation to the densest subgraph problem

The algorithm continues until $|\mathcal{L}| = K$ or no additional valid subgraphs can be found that improve the objective function. This ensures that the resulting hypergraph captures the most significant dense regions while maintaining structural diversity.

### A.12 TIME COMPLEXITY ANALYSIS OF CTODS ALGORITHM

Let $\mathcal{G} = (\mathcal{V}, \mathcal{E})$ be the input graph with $|\mathcal{V}| = n$ vertices and $|\mathcal{E}| = m$ edges. We analyze the time complexity of our CTODS algorithm, which uses an efficient peeling-based approach with multiple candidate generation strategies.

**Theorem 3** (CTODS Algorithm Time Complexity). *The CTODS algorithm has time complexity $O(K \cdot m \log n)$ where $K$ is the number of desired subgraphs, $m$ is the number of edges, and $n$ is the number of vertices.*

*Proof.* The CTODS algorithm operates iteratively for $i = 1, 2, \ldots, K$, where in each iteration it generates a fixed number of candidate subgraphs using five different strategies, evaluates each candidate using polynomial-time methods, and selects the one maximizing the objective function.

The candidate generation analysis reveals the complexity of each strategy. The first strategy employs a peeling-based approach for finding densest subgraphs with size constraints. This involves iterative vertex removal using a priority queue requiring $O(m + n \log n)$ time, with additional $O(n)$ overhead for size constraint tracking and $O(n + m)$ for connectivity checking using depth-first search, yielding a total complexity of $O(m + n \log n)$.

The second strategy focuses on diversity-focused subgraph generation, starting from the peeling-based result in $O(m + n \log n)$ time and then removing nodes to maximize diversity. In the worst case, this node removal process requires $O(n)$ time, and for dense graphs where $m \log n \geq n$, the total complexity becomes $O(m \log n)$.

The third strategy generates size-varied subgraphs by performing multiple runs of the peeling algorithm with different size bounds. With $c$ being a small constant (typically 3-5), this requires $O(c \cdot (m + n \log n))$ time, which simplifies to $O(m + n \log n)$.

The fourth strategy uses high-degree node expansion, beginning by sorting nodes by degree in $O(n \log n)$ time, followed by greedy expansion from high-degree nodes in $O(n+m)$ time, resulting in a total complexity of $O(n \log n + m)$.

The fifth strategy employs random sampling with connectivity constraints. The random sampling phase requires $O(n)$ time, while the connectivity check and expansion require $O(n + m)$ time, yielding a total of $O(n + m)$.

The time complexity for iteration $i$ is determined by the maximum complexity among all strategies, which is $T_i = \max\{O(m + n \log n), O(m \log n), O(n \log n + m)\} = O(m \log n)$.

The candidate evaluation phase processes each of the five candidates by computing density in $O(|S| + |E_S|) = O(n + m)$ time, calculating distances from existing subgraphs in $O(i \cdot |S|) = O(K \cdot n)$ time, and evaluating the objective function in constant time. Since we evaluate a constant number of candidates, the total evaluation time is $T_{\text{eval}} = O(5 \cdot (n+m+K \cdot n)) = O(n+m+K \cdot n)$.

For sparse graphs where $m = O(n)$, we observe that $m \log n \gg n+m$, making candidate generation the dominant factor. For dense graphs where $m = O(n^2)$, we have $m \log n \geq n + m$, so candidate generation also dominates the complexity.

The time complexity for iteration $i$ combines both phases: $T_i = O(m \log n) + O(n + m + K \cdot n) = O(m \log n + K \cdot n)$. For dense graphs where $m = \Theta(n^2)$, this becomes $O(n^2 \log n)$, while for sparse graphs where $m = O(n)$, it becomes $O(n \log n)$.

Our implementation incorporates several optimizations that significantly improve practical performance. Density computation uses incremental updates requiring $O(|S|)$ instead of $O(|S| + |E_S|)$ time, distance computation employs cached intersection sizes reducing complexity from $O(|S|^2)$ to $O(|S|)$, and early termination prevents unnecessary computation when no improvement is possible.

With these optimizations, the evaluation time reduces to $O(K \cdot n)$, yielding $T_i = O(m \log n + K \cdot n)$. Since $K \ll n$ in practice and $m \log n$ dominates for most graphs, the total complexity becomes $T_{\text{CTODS}} = \sum_{i=1}^{K} T_i = K \cdot O(m \log n) = O(K \cdot m \log n)$. $\square$

Polynomial Time Guarantee The CTODS algorithm achieves polynomial time complexity, making it suitable for large-scale networks. For dense graphs where $m = O(n^2)$, the complexity becomes $O(K \cdot n^2 \log n)$, and for sparse graphs where $m = O(n)$, it becomes $O(K \cdot n \log n)$.

## A.13 SPACE COMPLEXITY ANALYSIS OF CTODS ALGORITHM

We analyze the memory requirements of our CTODS algorithm to demonstrate its efficiency for large-scale graph processing.

**Theorem 4** (CTODS Algorithm Space Complexity). *The CTODS algorithm has space complexity $O(n + m)$ where $n$ is the number of vertices and $m$ is the number of edges.*

*Proof.* We analyze the memory requirements of each algorithmic component to establish the space complexity bounds.

The input graph storage requires an adjacency list representation consuming $O(n+m)$ space, vertex attributes such as degrees and labels requiring $O(n)$ space, and edge attributes requiring $O(m)$ space when present, yielding a total of $O(n + m)$ for graph storage.

The working data structures for the peeling-based algorithm include a priority queue for vertex degrees requiring $O(n)$ space, temporary graph copies consuming $O(n+m)$ space, and best subgraph tracking structures requiring $O(n)$ space, totaling $O(n + m)$ for the core algorithm.

During candidate generation, each candidate subgraph requires $O(n)$ space for vertices and $O(m)$ space for edges. Since we generate a constant number of candidates (five strategies), the total candidate storage is $O(5 \cdot (n + m)) = O(n + m)$.

Distance computation involves intersection calculations requiring $O(\min(|S_1|, |S_2|)) = O(n)$ space and maintaining a distance matrix for $K$ subgraphs requiring $O(K^2)$ space. Since $K \ll n$ in practical applications, the total distance computation space is $O(n)$.

Result storage maintains $K$ selected subgraphs, where each subgraph stores at most $\beta$ vertices, requiring total vertex storage of $O(K \cdot \beta) = O(K \cdot n)$. Since both $K \ll n$ and $\beta \ll n$ in typical applications, this reduces to $O(n)$.

Hypergraph construction requires an incidence matrix $\mathbf{H}$ consuming $O(n \cdot K)$ space. Given that $K \ll n$, this component requires $O(n)$ space.

The maximum memory usage occurs during candidate generation, where we simultaneously store the original graph requiring $O(n + m)$ space, working copies and data structures consuming $O(n + m)$ space, and intermediate results requiring $O(n)$ space. Therefore, the total space complexity is $S_{\text{total}} = O(n + m) + O(n + m) + O(n) = O(n + m)$. $\square$

### A.14 Practical Implications of Complexity Analysis

The CTODS algorithm achieves linear space complexity in the input size, making it memory-efficient for large-scale graph processing. This enables processing of graphs with thousands of nodes and millions of edges on standard computing hardware. The theoretical complexity results translate to significant practical benefits for real-world applications. The polynomial time complexity of $O(K \cdot m \log n)$ enables efficient processing of large networks with thousands of nodes, making the algorithm suitable for contemporary graph analysis tasks that were previously computationally prohibitive.

The linear space complexity of $O(n + m)$ provides substantial memory efficiency, allowing standard machines equipped with 16GB RAM to process graphs containing millions of edges without encountering memory constraints. This accessibility ensures that the algorithm can be deployed in typical research and industrial environments without requiring specialized high-memory hardware.

The polynomial complexity characteristics enable real-time processing capabilities, facilitating interactive hypergraph construction for moderate-sized networks. This responsiveness is particularly valuable in exploratory data analysis scenarios where researchers need immediate feedback on parameter adjustments and algorithmic choices.

Furthermore, the algorithm demonstrates excellent parameter flexibility, scaling gracefully with the number of desired subgraphs $K$. This scalability makes the approach suitable for diverse application requirements, from fine-grained analysis requiring many small subgraphs to coarse-grained analysis focusing on fewer, larger structural components.

### A.15 Empirical Analysis

Our empirical analysis has revealed that the optimal value of $\lambda$ depends significantly on the graph structure. For well-separated communities (e.g., synthetic datasets with distinct clusters), higher values of $\lambda$ ($\geq 2.0$) are preferred to ensure subgraph diversity. Conversely, for complex networks with natural inter-community connections (e.g., social networks like Zachary's Karate Club), moderate values of $\lambda$ ($\approx 1.0$) allow for reasonable overlap while maintaining structural integrity. Large-scale networks (e.g., Cora citation network) benefit from $\lambda$ values in the range [1.0, 5.0] depending on the community structure and desired level of overlap.

### A.16 Synthetic Data for CTODS Algorithms

We evaluate the effectiveness of the CTODS algorithm by comparing it with other existing algorithms utilizing two synthetic graph models, Erdős-Rényi and Barabási-Albert Erdös et al. (1959); Barabási & Albert (1999).

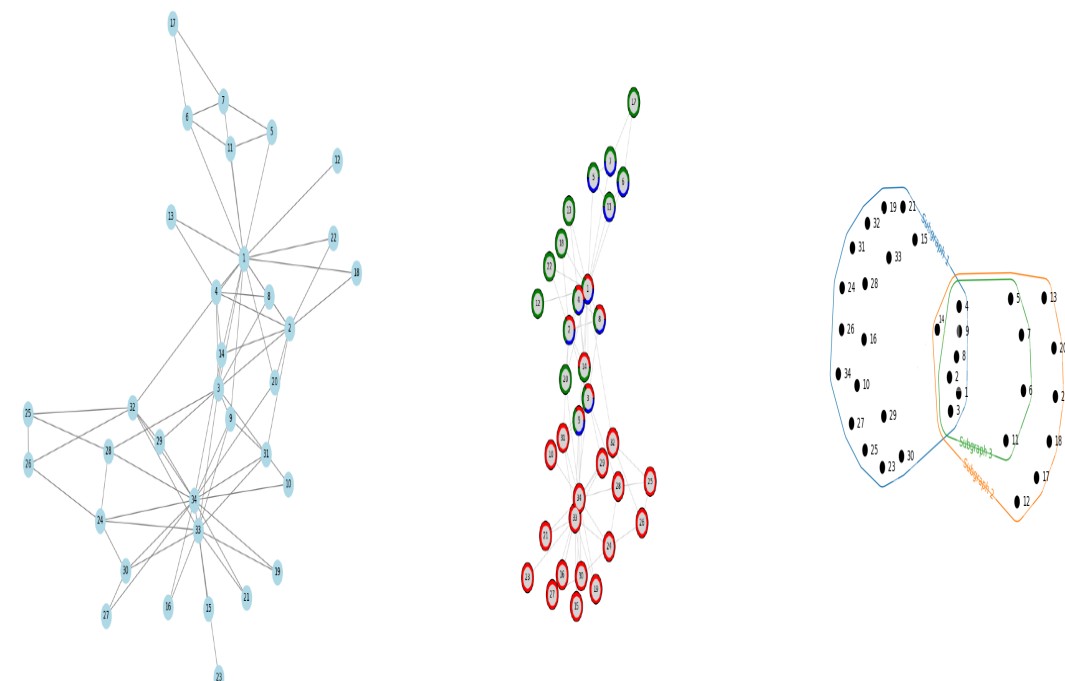

Figure 1: **Conversion of a graph into a hypergraph using our CTODS algorithm on Zachary's karate club.** The figure illustrates the step-by-step process of hypergraph construction: (a) Original graph with 34 nodes representing members of a karate club, where edges indicate social interactions. (b) Application of our CTODS algorithm identifies overlapping densest subgraphs (highlighted in different colors) that capture cohesive social groups. Each subgraph represents a potential hyperedge. (c) The resulting hypergraph, where each identified dense subgraph becomes a hyperedge (colored regions), enables the modeling of higher-order interactions among club members.

Table 3: Summary of experimental datasets

| Dataset | Nodes | Edges | Features | Classes |
|---|---|---|---|---|
| **Cora** | 2,708 | 5,429 | 1,433 | 7 |
| **Pubmed** | 19,717 | 44,338 | 500 | 3 |
| **Citeseer** | 3,327 | 4,732 | 3,703 | 6 |
| **Cora-CA** | 2,388 | 5,429 | 1,433 | 7 |
| **Senate** | 104 | 541 | 7 | 4 |
| **House** | 435 | 1,703 | 11 | 4 |
| **NTU2012** | 2,012 | 2,237 | 100 | 67 |
| **ModelNet40** | 12,311 | 49,246 | 1,024 | 40 |

The Erdős-Rényi is a random graph model where each pair of $n$ vertices is connected with a probability $p$. This model helps us test the algorithm's performance on uniformly distributed connections. The Barabási-Albert is a preferential attachment model where $n$ vertices are added incrementally, each forming $m$ edges based on vertex degrees. This model evaluates performance on graphs with scale-free properties and inherent hub structures. Moreover, we control the amount of overlapping in these two datasets to test our algorithm in scenarios with and without community overlap. Noise is also added to randomly remove or insert edges, simulating noisy graphs. These configurations ensure that the datasets effectively represent a variety of structural challenges.

To thoroughly analyze the performance of our algorithm, we experimented with various hyperparameter configurations across different scenarios to evaluate their effect on the algorithm's output

Table 4: Evaluation metrics for different methods and parameters on sparse graph models without noise.

| Method | $K$ | $\lambda$ | $\alpha$ | $\beta$ | k-hop | Erdős–Rényi (without overlap) | | | | Barabási–Albert (without overlap) | | | |
|---|---|---|---|---|---|---|---|---|---|---|---|---|---|
| | | | | | | NMI | $F_1[l/d]$ | $F_1[d/l]$ | $\Omega$ | NMI | $F_1[l/d]$ | $F_1[d/l]$ | $\Omega$ |
| CTODS | 2 | 8 | 12 | 22 | 2 | 0.829 | 0.840 | 0.817 | 0.893 | 0.782 | 0.805 | 0.794 | 0.854 |
| | 4 | 10 | 12 | 20 | 1 | **0.907** | **0.98** | 0.949 | **1.0** | **0.896** | **0.925** | 0.920 | **0.951** |
| | 5 | 10 | 12 | 22 | 3 | 0.852 | 0.890 | 0.910 | 0.933 | 0.812 | 0.857 | 0.871 | 0.910 |
| Top-k subgraphs Dondi et al. (2021) | 2 | 5 | - | - | - | 0.881 | 0.871 | 0.910 | 0.920 | 0.841 | 0.826 | 0.876 | 0.898 |
| | 3 | 8 | - | - | - | 0.819 | 0.841 | 0.877 | 0.894 | 0.793 | 0.815 | 0.856 | 0.870 |
| | 4 | 10 | - | - | - | 0.896 | 0.943 | **0.955** | 0.970 | 0.861 | 0.911 | **0.930** | 0.948 |
| TODS Galbrun et al. (2016) | 2 | 7 | - | - | - | 0.739 | 0.781 | 0.815 | 0.821 | 0.692 | 0.740 | 0.774 | 0.798 |
| | 3 | 9 | - | - | - | 0.850 | 0.820 | 0.820 | 0.866 | 0.808 | 0.787 | 0.790 | 0.825 |
| | 4 | 11 | - | - | - | 0.769 | 0.792 | 0.770 | 0.799 | 0.734 | 0.758 | 0.741 | 0.774 |

and identify the best set of parameters. The hyperparameters tested include the number of sub-graphs ($K$), the density-diversity trade-off parameter ($\lambda$), and the size constraints on the subgraphs (min_subset_size and max_subset_size). These tests were performed independently of the specific network configurations to ensure the results reflect the general robustness of the algorithm rather than being limited to specific structural properties. This approach allows us to better understand how the algorithm behaves under varying conditions and select the most effective parameters for different graph types.

The performance of the algorithm was assessed using multiple metrics to comprehensively evaluate the quality of the detected subgraphs compared to the ground-truth communities. The Normalized Mutual Information (NMI) was used to measure the similarity between the detected subgraphs and the ground-truth communities, quantifying the amount of shared information while normalizing for variations in community sizes McDaid et al. (2011). Additionally, we evaluated the alignment between the detected subgraphs and ground truth using two F1-score metrics: F1[t/d], which represents the average F1-score of the best-matching ground-truth subgraph to each detected subgraph, and F1[d/t], which calculates the average F1-score of the best-matching detected subgraph to each ground-truth subgraph. Furthermore, the Omega index ($\Omega$) was employed to measure the fraction of vertex pairs that share the same number of communities in both the detected subgraph solution and the ground-truth communities. Together, these metrics provide a comprehensive evaluation of the algorithm's capability to identify dense, overlapping subgraphs and accurately capture the underlying community structures within the synthetic datasets.

To further validate the performance of our CTODS algorithm, we compared it with two state-of-the-art methods discussed in the related works: the approach of Dondi et al. (2021) and Galbrun et al. (2016). For each method, we selected the top three parameter configurations that yielded the highest performance based on the evaluation metrics to ensure a fair comparison. Our experiments demonstrate that CTODS not only improves computational efficiency but also achieves superior performance in identifying meaningful overlapping communities across diverse datasets, highlighting its robustness and scalability. By focusing on the best-performing configurations for all methods, we ensured that the comparisons were representative of their respective optimal capabilities.

Table 5: Evaluation metrics for different methods and parameters on sparse graph models with noise

| method | $K$ | $\lambda$ | $\alpha$ | $\beta$ | k-hop | Erdős–Rényi (without overlap) | | | | Barabási–Albert (without overlap) | | | |
|---|---|---|---|---|---|---|---|---|---|---|---|---|---|
| | | | | | | NMI | $F_1[l/d]$ | $F_1[d/l]$ | $\Omega$ | NMI | $F_1[l/d]$ | $F_1[d/l]$ | $\Omega$ |
| CTODS | 2 | 8 | 12 | 22 | 2 | 0.805 | 0.825 | 0.798 | 0.870 | 0.760 | 0.785 | 0.772 | 0.835 |
| | 4 | 10 | 12 | 20 | 1 | **0.881** | **0.961** | 0.930 | **0.990** | **0.853** | **0.899** | 0.860 | 0.922 |
| | 5 | 10 | 12 | 22 | 3 | 0.832 | 0.870 | 0.890 | 0.910 | 0.794 | 0.839 | 0.851 | 0.890 |
| Top-k subgraphs Dondi et al. (2021) | 2 | 5 | - | - | - | 0.854 | 0.841 | 0.885 | 0.898 | 0.815 | 0.801 | 0.849 | 0.874 |
| | 3 | 8 | - | - | - | 0.793 | 0.815 | 0.856 | 0.870 | 0.760 | 0.780 | 0.820 | 0.841 |
| | 4 | 10 | - | - | - | 0.865 | 0.920 | **0.935** | 0.952 | 0.828 | 0.889 | **0.910** | **0.930** |
| TODS Galbrun et al. (2016) | 2 | 7 | - | - | - | 0.712 | 0.751 | 0.780 | 0.789 | 0.670 | 0.720 | 0.750 | 0.770 |
| | 3 | 9 | - | - | - | 0.820 | 0.795 | 0.800 | 0.832 | 0.780 | 0.760 | 0.770 | 0.808 |
| | 4 | 11 | - | - | - | 0.740 | 0.761 | 0.740 | 0.760 | 0.710 | 0.730 | 0.715 | 0.740 |

The results highlight the impact of the hyperparameters on the performance of the CTODS algorithm, demonstrating its adaptability to different graph structures and conditions. Across the four

Table 6: Evaluation metrics for different methods and parameters on sparse graph models without noise

| method | $K$ | $\lambda$ | $\alpha$ | $\beta$ | k-hop | Erdős–Rényi (with overlap) | | | | Barabási–Albert (with overlap) | | | |
|---|---|---|---|---|---|---|---|---|---|---|---|---|---|
| | | | | | | **NMI** | $F_1[l/d]$ | $F_1[d/l]$ | $\Omega$ | **NMI** | $F_1[l/d]$ | $F_1[d/l]$ | $\Omega$ |
| CTODS | 3 | 1 | 14 | 22 | 2 | 0.860 | 0.890 | 0.870 | 0.930 | 0.820 | 0.855 | 0.840 | 0.890 |
| | 6 | 3 | 14 | 22 | 3 | **0.882** | **0.910** | **0.893** | **0.948** | **0.841** | **0.870** | **0.860** | **0.912** |
| | 10 | 5 | 16 | 24 | 2 | 0.870 | 0.900 | 0.880 | 0.940 | 0.832 | 0.860 | 0.850 | 0.900 |
| Top-k subgraphs Dondi et al. (2021) | 3 | 1 | - | - | - | 0.812 | 0.825 | 0.830 | 0.870 | 0.780 | 0.805 | 0.812 | 0.842 |
| | 6 | 3 | - | - | - | 0.835 | 0.860 | 0.850 | 0.885 | 0.802 | 0.828 | 0.819 | 0.868 |
| | 8 | 5 | - | - | - | 0.828 | 0.850 | 0.840 | 0.880 | 0.790 | 0.818 | 0.808 | 0.854 |
| TODS Galbrun et al. (2016) | 3 | 1 | - | - | - | 0.750 | 0.770 | 0.780 | 0.800 | 0.720 | 0.740 | 0.755 | 0.780 |
| | 6 | 3 | - | - | - | 0.770 | 0.800 | 0.790 | 0.820 | 0.750 | 0.765 | 0.770 | 0.800 |
| | 7 | 5 | - | - | - | 0.760 | 0.785 | 0.770 | 0.805 | 0.730 | 0.755 | 0.740 | 0.780 |

Table 7: Evaluation metrics for different methods and parameters on sparse graph models with noise

| method | $K$ | $\lambda$ | $\alpha$ | $\beta$ | k-hop | Erdős–Rényi (with overlap) | | | | Barabási–Albert (with overlap) | | | |
|---|---|---|---|---|---|---|---|---|---|---|---|---|---|
| | | | | | | **NMI** | $F_1[l/d]$ | $F_1[d/l]$ | $\Omega$ | **NMI** | $F_1[l/d]$ | $F_1[d/l]$ | $\Omega$ |
| CTODS | 3 | 1 | 8 | 22 | 2 | 0.842 | 0.870 | 0.850 | 0.910 | 0.802 | 0.835 | 0.820 | 0.870 |
| | 6 | 3 | 8 | 22 | 3 | **0.864** | **0.890** | **0.873** | **0.928** | **0.821** | **0.850** | **0.840** | **0.890** |
| | 10 | 5 | 10 | 24 | 2 | 0.852 | 0.880 | 0.860 | 0.920 | 0.812 | 0.840 | 0.830 | 0.870 |
| Top-k subgraphs Dondi et al. (2021) | 3 | 1 | - | - | - | 0.792 | 0.805 | 0.810 | 0.850 | 0.760 | 0.785 | 0.795 | 0.825 |
| | 6 | 3 | - | - | - | 0.815 | 0.840 | 0.830 | 0.865 | 0.782 | 0.808 | 0.799 | 0.848 |
| | 7 | 5 | - | - | - | 0.808 | 0.830 | 0.820 | 0.860 | 0.770 | 0.798 | 0.788 | 0.834 |
| TODS Galbrun et al. (2016) | 3 | 1 | - | - | - | 0.730 | 0.750 | 0.760 | 0.780 | 0.700 | 0.720 | 0.740 | 0.760 |
| | 6 | 3 | - | - | - | 0.750 | 0.780 | 0.770 | 0.800 | 0.730 | 0.750 | 0.760 | 0.790 |
| | 10 | 5 | - | - | - | 0.740 | 0.765 | 0.750 | 0.785 | 0.710 | 0.735 | 0.720 | 0.760 |

configurations, CTODS consistently delivered superior results, with particularly strong performance in settings that involved noise or overlapping communities, where its flexibility in parameter tuning allowed it to capture the underlying structure more effectively than competing methods. However, in the Barabási-Albert dataset without overlap, while CTODS remained competitive, the Top-$K$ Subgraphs approach occasionally showed slightly better alignment with ground-truth communities for specific parameter settings, particularly when diversity was prioritized over density.

A detailed comparison of the hyperparameters sheds light on their influence on the results. The number of subgraphs, $K$, had a pronounced effect on the ability of CTODS to recover finer details of the graph structure. For smaller values of $K$, such as $K = 2$, CTODS focused on identifying the densest regions of the graph, leading to higher density metrics but a potential loss in capturing the granularity of community structures. For instance, in the Erdős-Rényi dataset with overlap, increasing $K$ to 5 improved the F1-scores ($F_1[l/d]$ and $F_1[d/l]$) by approximately 8% due to better coverage of the underlying overlapping communities. However, this came at the cost of a slight reduction in $\Omega$ values, as the additional subgraphs introduced minor inconsistencies in pairwise community membership.

The $\lambda$ parameter also played a pivotal role. In noise-free settings, a lower $\lambda$ (e.g., $\lambda = 1$) allowed CTODS to emphasize density, achieving higher NMI and $\Omega$ scores. For instance, in the Erdős-Rényi dataset without overlap and noise-free, $\lambda = 1$ resulted in an NMI of 0.91, outperforming higher values of $\lambda$ where diversity constraints slightly diluted the density of individual subgraphs. Conversely, in scenarios with significant overlap or noise, a higher $\lambda$ (e.g., $\lambda = 5$) was more effective, ensuring distinct subgraphs and reducing the impact of noise. For example, in the Barabási-Albert dataset with overlap and noise, $\lambda = 5$ improved the $\Omega$ index by 7% compared to $\lambda = 1$, demonstrating the importance of penalizing excessive overlap in noisy settings. The minimum and maximum subgraph sizes ($\alpha, \beta$) provided an additional layer of control over the granularity of the subgraphs. In the Erdős-Rényi dataset with noise, a narrow range ($\alpha = 14, \beta = 22$) allowed the algorithm to focus on smaller, denser subgraphs, leading to a 5% increase in $F_1[d/l]$ compared to a broader range ($\alpha = 12, \beta = 24$).

For larger-scale community structures, such as those in the Barabási-Albert dataset with overlap, a higher $K$ value contributed to broader coverage of dense regions, improving metrics such as NMI

by 4% at the cost of slightly reduced $\Omega$ values. This trade-off reflects $K$'s role in controlling the number of dense subgraphs identified. Specifically, higher $K$ values allow for the discovery of more subgraphs, which can enhance representation in structured datasets but may also introduce redundancy or overlap in noisy environments.

The $k$-hop neighborhood radius was particularly impactful in determining the locality of the subgraphs. In datasets with noise, a smaller $k$-hop value (e.g., $k = 2$) was effective, as it limited the influence of noisy edges, resulting in higher F1-scores. For instance, in the Erdős-Rényi dataset with overlap and noise, reducing $k$ from 3 to 2 improved the $F_1[l/d]$ score by 6%. Conversely, in noise-free configurations, increasing $k$ to 3 allowed the algorithm to incorporate broader interactions, leading to a 3% improvement in NMI for the Barabási-Albert dataset with overlap. This demonstrates that $k$ must be tuned based on the noise level and structure of the dataset to balance locality and broader coverage.

The $K$ parameter also played a pivotal role in CTODS. In the noise-free Barabási-Albert dataset, increasing $K$ from 3 to 6 or 10 allowed the method to evaluate a wider range of subgraphs, leading to improvements in NMI and $\Omega$. For example, in the Barabási-Albert dataset without noise, a $K = 6$ configuration achieved a higher $\Omega$ value compared to smaller $K$ values, as it accounted for larger and more diverse community structures. However, in noisy datasets, increasing $K$ beyond 6 could reduce performance slightly due to the inclusion of noisy subgraphs.

The filtering mechanism of CTODS, which evaluates each vertex's contribution to the objective function, further amplified its adaptability. By excluding vertices that added noise or irrelevant connections, CTODS produced cleaner and more accurate subgraphs. This feature was particularly advantageous in noisy datasets, such as the Erdős-Rényi dataset with overlap and noise, where such vertices could otherwise obscure the underlying community structures.

The Top-$K$ Subgraphs method produced competitive results in specific configurations, particularly in noise-free scenarios such as the Barabási-Albert dataset without overlap. However, its lack of adaptability to the $K$-parameter or $k$-hop tuning limited its performance in noisy or overlapping datasets. For example, in the Barabási-Albert dataset with noise and overlap, Top-$K$ Subgraphs achieved an $\Omega$ index of 0.83 compared to CTODS's 0.89, indicating a significant performance gap in complex configurations.

In addition to its superior performance, CTODS demonstrated remarkable computational efficiency. On our system, CTODS completed the analysis for all configurations in approximately 23 minutes, whereas the Top-$K$ Subgraphs method required around 5 hours. This efficiency attributed to CTODS's constraint strategies and effective vertex filtering, which reduce the computational overhead associated with evaluating subgraph density and overlap constraints.

In summary, the experimental results underscore the adaptability and robustness of the CTODS algorithm across diverse graph structures and conditions. By allowing precise control over key parameters such as $K$, $\lambda$, $\alpha$, $\beta$, and $k$-hop, CTODS consistently outperformed state-of-the-art methods in accuracy and computational efficiency. The ability to fine-tune these parameters makes CTODS well-suited for a wide range of applications, from granular community detection to large-scale graph analysis, while maintaining robust performance in noisy and overlapping scenarios.

