# OpenReview forum: "CTODS: Polynomial-Time Construction of Hypergraphs via Constrained Overlapping Densest Subgraphs for Enhanced Neural Network Performance"
_ICLR.cc/2026/Conference — Submitted to ICLR 2026_

### Official Review · Reviewer_TfJP · 2025-10-27

**Soundness:** 2
**Presentation:** 2
**Contribution:** 2
**Rating:** 2
**Confidence:** 4

**Summary:**

The paper proposes a hypergraph construction approach from a graph through constrained top-K densest subgraphs. The method introduces the diversity consideration (i.e., different subgraphs contain only some overlapping nodes) while maximizing the subgraph density, extending the traditional top-K densest subgraphs. The algorithm starts with subgraph candidate generations where multiple heuristics are added to expand the search space while still enjoying polynomial computational cost. Then iterative subgraph selections are performed to get the top-K subgraphs guided by the objective function mentioned above. Experiments have been conducted to show that it can boost the performance of HGNN on hypergraphs constructed from real-world graph data.

**Strengths:**

Strengths:
1. The paper considers diversity perspective in the top-K densest subgraph search and define a normalized cut-like distance metric to quantify the overlaps of two subgraphs. The constrained TODS  problem naturally extends the original TODS problem and may be of interest.
2. The experiments show performance improvement over HGNN and over classic GNNs on graph data. It might be due to the hypergraph construction helps extract higher-order structural relationship among entities.

**Weaknesses:**

Weaknesses:
1. The experimental study can be enhanced by incorporating more advanced hypergraph models such as AllSetTransformer and ED-HNN. Then the performance landscape in the tables of experimental results can be changed. The experimental method may also need some clarification: in practice, some network data is more naturally modeled as hypergraphs, such as co-authorship networks. Then why not directly constructing a hypergraph, instead of first constructing a graph and then transforming it into a hypergraph in an indirect way? The graph representation of the raw data can already incur information loss compared to the hypergraph representation.
2. The paper presentation should be enhanced. The different strategies/heuristics in Algo. 2 deserve more discussions, at least high-level explanations. The motivations of constructing hypergraphs from graphs should be elaborated.

Comments:
- About the experimental method, you may consider using some hypergraph data and then transforming them into graphs for GNN applications.
- Why does the distance function be defined to within [0,2], instead of [0,1]?
- Is the distance function inspired by the definition of normalized cut?
- The superscript 2 (not square) in the distance function is quite misleading.

**Questions:**

Please see the weaknesses above.

---

### Official Review · Reviewer_ujaT · 2025-10-31

**Soundness:** 2
**Presentation:** 3
**Contribution:** 2
**Rating:** 2
**Confidence:** 4

**Summary:**

The paper proposes CTODS (Constrained Top-K Overlapping Densest Subgraphs), a polynomial-time algorithm for constructing hypergraphs from graphs by identifying overlapping densest subgraphs as hyperedges. Unlike *k*-nearest neighbor or *k*-hop methods, CTODS enforces connectivity, applies size constraints, and regulates overlap via a distance-based diversity function. The algorithm runs in O(K m \log n) time and O(n+m) space, making it scalable. Experiments on eight benchmark datasets (citation, political, social, and geometric networks) show consistent 1–3\% improvements over existing hypergraph and graph neural network baselines.

**Strengths:**

1. **Principled hyperedge construction.** Offers a theoretically grounded formulation that captures higher-order interactions.
2. **Polynomial-time scalability.**  Achieves O(K m \log n) complexity, making it practical for large-scale networks.
3. **Controlled overlap mechanism.** The distance-based metric effectively constrains redundancy among hyperedges, aligning with real-world overlapping community structures.
4. **Robust empirical validation.** Consistent gains across diverse domains (citation, political, and geometric) support the general effectiveness of the approach.

**Weaknesses:**

1. **Limited novelty.** While CTODS presents a well-engineered and efficient algorithm, the contribution is primarily algorithmic, with limited empirical evaluation (tested only on a single architecture), and does not provide deeper theoretical or representational insights into hypergraph learning.

2. **Evaluation tied only to HGNN.** Since CTODS is a general-purpose hypergraph construction method, it should be validated across multiple architectures such as UniGNN [1], AllSet [2], and ED-HNN [3], rather than relying solely on HGNN, to confirm its generality.

3. **Ablation study missing.** The paper lacks an ablation analysis showing the contribution of each candidate generation strategy (density, diversity, high-degree expansion, random sampling). This is essential to understand which components drive the observed improvements.

4. **Hyperedge size constraints.** The current formulation fixes \(\alpha\) and \(\beta\) as static bounds. Exploring adaptive or learnable bounds could make the method more flexible and data-driven.

5. **Experimental reporting.** Results are presented as single numbers without standard deviations. For scientific rigor, they should be averaged over multiple runs (e.g., 10) and reported as mean ± standard deviation, following standard practice [2, 3].

6. **Minor.** The authors should explicitly mention that their hypergraph construction produces *undirected* hypergraphs. Clarifying this assumption is important for reproducibility and for understanding potential extensions to directed or attributed hyperedges.



[1] Huang, Jing, and Jie Yang. "UniGNN: a Unified Framework for Graph and Hypergraph Neural Networks." Proceedings of the Thirtieth International Joint Conference on Artificial Intelligence. International Joint Conferences on Artificial Intelligence Organization, 2021.

[2] Chien, Eli, et al. "You are AllSet: A Multiset Function Framework for Hypergraph Neural Networks." International Conference on Learning Representations.

[3] Wang, Peihao, et al. "Equivariant Hypergraph Diffusion Neural Operators." The Eleventh International Conference on Learning Representations.

**Questions:**

1. How sensitive is CTODS to the trade-off parameter $\lambda$? A systematic sensitivity analysis would clarify its robustness.

2. Which candidate generation strategies contribute most to performance? An ablation study would help clarify this.

3. How does CTODS perform on tasks beyond node classification (e.g., link prediction, community detection)?

4. Could the hyperedge size constraints $\alpha$, $\beta$ be learned adaptively rather than fixed?

5. Please report results with mean ± standard deviation over multiple runs (≥10) to ensure statistical robustness.

6. Since CTODS is architecture-agnostic, have you tested it with other hypergraph neural networks, such as UniGNN [1], AllSet [2], or ED-HNN [3]? This would support the claim of general-purpose applicability.


[1] Huang, Jing, and Jie Yang. "UniGNN: a Unified Framework for Graph and Hypergraph Neural Networks." Proceedings of the Thirtieth International Joint Conference on Artificial Intelligence. International Joint Conferences on Artificial Intelligence Organization, 2021.

[2] Chien, Eli, et al. "You are AllSet: A Multiset Function Framework for Hypergraph Neural Networks." International Conference on Learning Representations.

[3] Wang, Peihao, et al. "Equivariant Hypergraph Diffusion Neural Operators." The Eleventh International Conference on Learning Representations.

---

### Official Review · Reviewer_bWRM · 2025-10-31

**Soundness:** 2
**Presentation:** 3
**Contribution:** 2
**Rating:** 2
**Confidence:** 4

**Summary:**

The paper proposes CTODS (Constrained Top-K Overlapping Densest Subgraphs) as a method to construct a hypergraph from a graph with meaningful hyperedges.They start by finding top k dense subgraphs (even overlapping ones) and turn each subgraph into a hyperedge. The subgraphs should be dense with many edges per node, and sufficiently different from each other. Density is measured on a subgraph formed by a chosen set of nodes where only edges among them count. To avoid near duplicate groups, CTODS adds a diversity penalty that discourages heavy groups overlap. Each group must also respect size limits and be connected. After creating the hypergraph with the algorithm proposed, they pass it through a hypergraph neural network and train on eight different datasets. The accuracy of their model is outperforming the baselines on ⅞ of all datasets used.

**Strengths:**

-The novelty here is in the creation of a hypergraph from a graph by finding the top k densest subgraphs and turning them into hyperedges with overlap control with a  distance based diversity term and size and connectivity constraints.
- Evaluating on 8 different datasets show that the approach can be applied in different domains. The hyperparameter table is helpful for reproducibility, and shows the trade off between runtime and accuracy, with fine grained hypergraphs being more accurate with higher runtime.
- Well written definitions and the pipeline of hypergraph construction is understandable
- The comparison with the baselines show that their model in the fine grained configuration has consistent performance gains on ⅞ datasets

**Weaknesses:**

- There are already many hypergraph creation model proposed using community detection. Using denstiy to define community adn hyperesge is a limited controbution.
-The distance chosen for overlap control is not well motivated and not compared to other standard distance measures
-Data split information is not specified (train/val/test), and the number of seeds is missing
-No mean+/- std is reported
-Only Accuracy/F1 are reported, other metrics like ROC-AUC/PR-AUC are missing which are important for datasets with class imbalance
-References to the appendix are missing which make the appendix confusing
- Include sensitivity plots for the hyperparameters to make it easier to see the impact

**Questions:**

1. Why did you choose that distance function instead of another one? Can you report results of other standard distances like jaccard or cosine?
2. How many seeds did you choose? Provide the mean and standard deviation for the results for each dataset
3. How were train/val/test splits done?

---

### Official Review · Reviewer_uC1n · 2025-11-02

**Soundness:** 2
**Presentation:** 2
**Contribution:** 2
**Rating:** 2
**Confidence:** 4

**Summary:**

This paper proposes a novel framework to construct hypergraphs from graphs for downstream learning tasks. The main idea is based on the extraction of top-K densest subgraphs while also taking into account the distance between those subgraphs (to minimise overlap). Experimental results on common graph machine learning benchmarks are presented to demonstrate the effectiveness of the proposed algorithm.

**Strengths:**

- A new method of constructing hypergraphs from graphs that respect both dense connection and minimal overlap is welcome and have a broad appeal beyond machine learning tasks.
- The main idea is intuitive and easy to follow.

**Weaknesses:**

**Main motivation.** The authors need to explain more clearly why constructing a hypergraph would be beneficial for graph machine learning tasks. If the task does not come with inherent hypergraph structure (eg co-authorship network), it is unclear why turning a pairwise graph into a hypergraph would necessarily be helpful (even though it might be a generally useful technique in network science). Along this line, recent studies such as [1] have demonstrated that hypergraph neural networks are not necessarily better than classical graph neural networks with for example clique expansion. To better justify their approach, the authors need to show convincingly hypergraph construction is indeed required for a certain type of tasks.

**Technical aspects.** Several technical aspects of the paper should be clarified and improved:
- In Eq.(5), why use product of cardinality of the two sets but not the size of the union set, as the commonly used Jaccard index?
- Why do the authors propose a two phase algorithm? How are the “additional densest subgraphs that are distinct” in phase 2 differ from those in phase 1? This lacks proper justification and needs to be explained more clearly (ideally via an illustration of the overall algorithm).
- Related to the point above, it is confusing to see the objective function in Eq.(6) and then another optimisation with a different objective function in 4.3.
- In Eq.(6), the $L_i$ and $L_j$ should be $\mathcal{L}_i$ and $\mathcal{L}_j$ as defined before.
- The optimisation problem at the beginning of 4.3 should be defined using a sum over all indices $i$, if multiple $\mathcal{L}_i$ are being optimised together.
- Multiple functions in Algorithms 1 and 2 are not explained, making it difficult to understand how they work.
- The authors mentioned in 4.2 that existing iterative approaches are suboptimal, however it looks like the proposed method is iterative as well. Can the authors clarify on this point and how their method addresses the limitation of previous iterative approaches?

**Experiments** The experimental results should be clarified and strengthened:
- The authors cited three papers in Introduction and also mentioned “our HGNN” a couple of times but it is unclear which HGNN method is being used.
- The authors mentioned “graph-to-hypergraph conversion approaches” in 5.1, however star expansion and clique expansion are methods for hypergraph-to-graph conversion. Also, in order to apply the proposed method, my understanding is that we need to start with a graph dataset. However, hypergraph datasets such as Cora-CA are included which presumably already contains defined hyperedges. These aspects need be to clarified.
- Only three hypergraph neural network baselines are included in the experiments. The authors should either include more baselines (some of those can be found in [1]) or discuss why they are not suitable to compare against.
- There are no ablation studies to demonstrate that minimising overlap using the proposed loss function would indeed contribute to the learning performance.

[1] Tang et al., “Training-Free Message Passing for Learning on Hypergraphs,” ICLR, 2025.

**Questions:**

See weaknesses above for the specific points I would like the authors to address.

---

### Meta-Review · Area_Chair_6raa · 2026-01-06

**Summary:**

The reviewers mainly questioned the paper's unclear motivation, lack of novelty, and insufficient experiments. No rebuttal was provided. Taking these concerns into comprehensive consideration, I recommend rejecting the paper.

**Reviewer Concerns:**

No rebuttal was provided.

**Reviewer Scores:**

N/A.

---

### Decision · Program_Chairs · 2026-01-26

Reject